

**Improving estimates of water resources in a semi-arid region by assimilating GRACE**
**data into the PCR-GLOBWB hydrological model**
N. Tangdamrongsub[1,2], S. C. Steele-Dunne[3], B. C. Gunter[1,4], P. G. Ditmar[1], E. H.
Sutanudjaja[5], Y. Sun[1], T. Xia[6], and Z. Wang[6,7]
[1] Department of Geoscience and Remote Sensing, Faculty of Civil Engineering and
Geosciences, Delft University of Technology, Delft, The Netherlands
[2] School of Engineering, Faculty of Engineering and Built Environment, The University of
Newcastle, Callaghan, New South Wales, Australia
[3] Department of Water Resources, Faculty of Civil Engineering and Geosciences, Delft
University of Technology, Delft, The Netherlands
[4] School of Aerospace Engineering, Georgia Institute of Technology, Atlanta, The United
States of America
[5] Department of Physical Geography, Faculty of Geosciences, Utrecht University, Utrecht,
The Netherlands
[6] Department of Hydraulic Engineering, Tsinghua University, Beijing 100084, China
[7] State Key Lab of Hydroscience and Engineering, Tsinghua University, Beijing 100084,
China
Correspondence to: N. Tangdamrongsub (Natthachet.tangdamrongsub@newcastle.edu.au)
**Abstract**
An accurate estimation of water resources dynamics is crucial for proper management of both
agriculture and the local ecology, particularly in semi-arid regions. Imperfections in model
physics, uncertainties in model land parameters and meteorological data, as well as the human
impact on land changes often limit the accuracy of hydrological models in estimating water
storages. To mitigate this problem, this study investigated the assimilation of Terrestrial
Water Storage (TWS) estimates derived from the Gravity Recovery And Climate Experiment
(GRACE) data using an Ensemble Kalman Filter (EnKF) approach. The region considered
was the Hexi Corridor of Northern China. The hydrological model used for the analysis was
PCR-GLOBWB, driven by satellite-based forcing data from April 2002 to December 2010. In
this study, EnKF 3D scheme, which accounts for the GRACE spatially-correlated errors, was
used. The correlated errors were propagated from the full error variance-covariance matrices
provided as a part of the GRACE data product. The impact of the GRACE Data Assimilation
(DA) scheme was evaluated in terms of the TWS, as well as individual hydrological storage
estimates. The capability of GRACE DA to adjust the storage level was apparent not only for
the entire TWS but also for the groundwater component, which had annual amplitude, phase,
and long-term trend estimates closer to the GRACE observations. This study also assessed the
benefits of taking into account correlations of errors in GRACE-based estimates. The
assessment was carried out by comparing the EnKF results, with and without taking into
account error correlations, with the in situ groundwater data from 5 well sites and the in situ
streamflow data from two river gauges. On average, the experiments showed that GRACE
DA improved the accuracy of groundwater storage estimates by as much as 25%. The



inclusion of error correlations provided an equal or greater improvement in the estimates. No
significant benefits of GRACE DA were observed in terms of streamflow estimates, which
reflect a limited spatial and temporal resolution of GRACE observations. Results from the 9-
year long GRACE DA study were used to assess the status of water resources over the Hexi
Corridor. Areally-averaged values revealed that TWS, soil moisture, and groundwater
storages over the region decreased with an average rate of approximately 0.2, 0.1, and 0.1
cm/yr in terms of equivalent water heights, respectively. A substantial decline in TWS
(approximately –0.4 cm/yr) was seen over the Shiyang River Basin in particular, and the
reduction mostly occurred in the groundwater layer. An investigation of the relationship
between water resources and agriculture suggested that groundwater consumption required to
maintain the growing period in this specific basin was likely the cause of the groundwater
depletion.

## 1. Introduction

Semi-arid regions can be broadly classified as areas on the boundaries of larger deserts,
receiving just enough annual precipitation (300 mm or less) to sustain a limited amount of
agriculture. Inefficient use of the limited amount of surface water can often lead to overuse of
groundwater resources and salinization of the soil. This can result in desertification, which not
only reduces the amount of production but also may have long-term effects on the local
ecology. Improving the water resources management of such regions requires accurate
knowledge of the hydrological processes involved.  For small areas, this can be partially
obtained through a network of in-situ measurement systems, such as meteorological stations,
river gauges, groundwater wells, evaporation trays, etc. (Dahlgren & Possling, 2007; Huo et
al., 2007; Kang et al., 2004; Ma et al., 2005; Du et al., 2014). However, as point
measurements, these observations are very local in scope.  A sensor at a point several
kilometres away may record significantly different values. For large scales (> 10,000 km$^2$),
such techniques are unlikely capable of delivering accurate results.
Two options for estimating the variations of the large-scale Terrestrial Water Storage (TWS)
of a particular region are: using observations from the Gravity Recovery And Climate
Experiment satellite mission (GRACE, Tapley et al., 2004) or utilizing a regional or global
hydrological model. A number of prior studies have reported on the potential of GRACE in
the estimation of snow water equivalent (Niu et al., 2007), groundwater (Döll et al., 2014),
and evapotranspiration (Long et al., 2014) in terms of temporal and spatial variability.
However, GRACE only provides the total column of the water storage at a monthly time scale
and large spatial scales (> 300 km). Furthermore, it is not possible to identify the contribution
of separate hydrological components to the TWS from GRACE data alone. On the other hand,
a hydrological model can be used to estimate the individual storage components at very high
spatial and temporal scale. The major drawback of the model is mainly the significant
uncertainties influenced by the quality of the model parameter calibration and the accuracy of
the meteorological input data.



Data Assimilation (DA) can be employed to combine the strengths of GRACE and
hydrological models while mitigating their respective weaknesses. A number of studies show
that GRACE DA can be used to improve the estimation of groundwater and streamflow
(Zaitchik et al., 2008; Tangdamrongsub et al., 2015), snow water equivalent (Forman et al.,
2012; Su et al., 2012), and as well as for evaluation of drought events (Houborg et al., 2012;
Li et al., 2012). Different temporal and spatial resolution of GRACE observations and
hydrological models require proper design of the DA scheme. Several DA schemes have been
developed to distribute GRACE observations into the model, which include assuming a
uniform observation value available every 10 days and updating the model every 10 days
(Zaitchik et al., 2008), using 5-day interpolated observations and updating the model every 5
days (Tangdamrongsub et al., 2015), using a monthly observation value and applying the
model update only at the end of the month (Eicker et al., 2014), and using a monthly
observation value and distributing the update as a daily increment (Forman et al., 2012).
Although all DA schemes are acceptable, the scheme proposed by Forman et al. (2012) is
advantageous because it does not require an interpolation of the observations and can reduce
the spurious jump caused by applying the update at the end of the month only. As such, a
scheme similar to (Forman et al., 2012) is used in this study. Spatial disaggregation is also
needed to reconcile the difference in horizontal resolution between the observations and the
model. Recent studies by Forman et al. (2013) and Eicker et al. (2014) suggested including
the GRACE variance-covariance error information in the spatial disaggregation step. Forman
et al. (2013) recommended using the finest observation resolution where observation error is
considered uncorrelated, while Eicker et al. (2014) proposed using 500-km GRACE spatial
resolution to mitigate the ill-posedeness of the error covariance matrices in the spatial domain.
In line with Forman et al. (2013) and Eicker et al. (2014), the assimilation scheme in this
study accounts for spatially correlated errors by using full error variance-covariance matrices
of GRACE data. This study will show that considering the GRACE error correlations leads to
an improvement of the state estimates.
The semi-arid region investigated in this study, the Hexi Corridor, is located between the
Gansu province of China and Mongolia (Fig. 1). In the past 50 years, over-utilization of the
groundwater has significantly reduced the groundwater supply and quality there. This has
resulted in increased soil salinization, to the extent that desertification has become a genuine
threat to both agriculture and the natural ecology (Wang et al., 2003). Due to a small size of
the Hexi Corridor area, the Signal-to-Noise Ratio (SNR) of the TWS variations estimated
there on the basis of GRACE data is much lower than in river basins considered in the
previous studies, e.g., Mississippi (Zaitchik et al., 2008), Rhine (Tangdamrongsub et al.,
2015), and Mackenzie (Forman et al., 2012). GRACE observations are assimilated into the
PCRaster Global Water Balance (PCR-GLOBWB; Van Beek et al., 2011; Sutanudjaja et al.,
2014; Wada et al., 2014) hydrological model over the Hexi Corridor. TWS is computed from
PCR-GLOBWB as the sum of all the hydrological components (soil moisture, groundwater,
surface water, inundated water, interception, and snow). The previous studies showed very
strong correlations of PCR-GLOBWB based estimates with GRACE observations in several
river basins (Wada et al., 2014; Tangdamrongsub et al., 2016). However, the performance of
PCR-GLOBWB has not been evaluated over the Hexi Corridor yet. In addition, the model has



not been incorporated into any GRACE DA scheme, making this study the first attempt to do
so.
The results of the GRACE DA approach are validated with independent in-situ and remote
sensing data. The groundwater storage (GWS) and streamflow estimates are validated with
the well and river stream gauge measurements, respectively. The precipitation from the
Tropical Rainfall Measuring Mission (TRMM; Huffman et al., 2007) and the Moderate
Resolution Imaging Spectroradiometer (MODIS) derived Normalized Difference Vegetation
Index (NDVI; Huete et al., 2002) are used to investigate the connection between agricultural
activity and the groundwater consumption in the area.
The main objective of this study is to investigate the added value of GRACE DA in the Hexi
Corridor. Approximately 9 years of data – between April 2002 and December 2010 – are
considered. The performance of the GRACE DA scheme is evaluated in terms of its impact
on the total terrestrial water storage as well as on the individual hydrological storage
estimates. The impact of taking into account correlation in GRACE errors is also assessed.
This will be shown to improve the storage estimates, particularly groundwater. Finally, results
from this 9-year long GRACE DA study are used to assess the status of water resources over
the Hexi Corridor. The connections between the water storage, especially groundwater, and
agriculture in the area are also presented and discussed.

## 2. Study region

The Hexi Corridor is a long and narrow area between the Qilian Mountain range and southern
Mongolia (Fig. 1a). The basin's elevation ranges from 5,200 m in the southern upstream area
(Qilian Mountains) to 900 m in the northern downstream zone (Inner Mongolia) (Fig. 1b).
The region is comprised of four typical inland arid and semi-arid regions: the Shiyang River
Basin (41,600 km$^2$), the Heihe River Basin (143,000 km$^2$), the Shule River Basin (157,000
km$^2$), and a Desert Region (152,445 km$^2$) (Geng and Wardlaw, 2013; Zhu et al., 2015).
Located next to the Gobi Desert, most parts of the region have a cold desert climate, where
precipitation is relatively low to sustain vegetation or crops. Approximately 60 to 80 % of the
annual rainfall is concentrated during the timeframe from June to September. The inland
rivers mainly originate from the Qilian Mountains and disappear after entering the
midstream/downstream plains and oases. As such, the southern part of the region is more
favourable for agriculture.
The four basins have distinct characteristics. First, the smallest river basin, Shiyang, has 8
main river streams, including the Xida and Xiying Rivers (Fig. 1c). The annual rainfall and
the mean temperature are approximately 250 mm and 5 °C (Fig. 2a, b), respectively. The
Shiyang River Basin is considered the wettest basin compared to the others, with relatively
high mean total renewable annual water resources of approximately 1.66 billion m$^3$ (Zheng et
al., 2013). However, a highly developed economy and population growth in the past decade
have resulted in a severe water resources overexploitation problem (Zheng et al., 2013). The
Heihe River Basin has a semi-arid climate and the mean daily temperature of ~6 °C (Fig. 2d).



The average annual rainfall is ~150 mm (Fig. 2c) with high heterogeneity both in temporal
and spatial distribution. The mean total annual available water resources are estimated at 3.7
billion m$^3$ (Hu, 2015). Similar to the Shiyang River Basin, increased water exploitation,
increasing population, and changing climate have aggravated the damage to the downstream
ecology. The Shule River Basin has an arid climate, the mean temperature there is around 4
°C (Fig. 2f), and the average annual rainfall is only approximately 98 mm (Fig. 2e).
Compared to the Shiyang River Basin, the Shule River Basin is approximately four times as
large in terms of surface area, but has similar mean total annual water resources, ~1.6 billion
m$^3$ (Hu, 2015). The district irrigation areas are mainly located in the middle of the Shule
River Basin. Agricultural water consumption accounts for more than 80% of the total water
use. Finally, the Desert Region has an extreme continental desert climate with an average
temperature of 8 °C, and the annual rainfall of ~130 mm. Extensive groundwater abstraction
was also observed over the region (Jiao et al., 2015).

**3. Hydrology model**
The global hydrological model PCR-GLOBWB (van Beek et al., 2011; Sutanudjaja et al.,
2016) simulates spatial and temporal continuous fields of fluxes and storages in various water
storage components (soil moisture, groundwater, surface water, inundated water, interception,
and snow). The model version used here (Sutanudjaja et al., 2016) has a spatial resolution of
30 arc minutes (approximately 50 km at the equator). It includes an interactive module
simulating water availability and water abstraction, including variations in irrigation and other
sectoral water demands (e.g. livestock, industrial and domestic water demands) and dynamic
allocation of available surface water and groundwater resources (see also, e.g., de Graaf et al.,
2014; Wada et al., 2014; Sutanudjaja et al., 2015). This feature determines how the demands
are allocated to the available water resources and where the return flows of unconsumed water
go. Modelling this feedback is essential, particularly in irrigated areas where water demand is
large. Details of the water demand calculation in PCR-GLOBWB can be found in Appendix
A.
Figure 3 illustrates the structure of PCR-GLOBWB model. The model includes 2 soil layers
($SM_{upp}$, $SM_{low}$), an underlying hydrologically active and replenishable groundwater
($GWS_{active}$) layer, a non-renewable groundwater ($GWS_{fossil}$) layer, as well as an interception,
surface water, and snow stores. The non-renewable groundwater is available for abstraction to
satisfy water demands once the overlying hydrologically active groundwater storage is
depleted. For soil, snow, inundated top water, and interception stores, an individual grid cell is
divided into sub-grids associated with different types of topography, vegetation phenology,
and soil properties, as well as land cover types. Specifically, there are 4 types of land covers
defined: short natural vegetation, tall natural vegetation, irrigated non-paddy field, and
irrigated paddy field. Soil components include the upper layer ($SM_{upp}$, 0 – 30 cm) and the
lower layer ($SM_{low}$, 30 – 150 cm). The snow component includes snow water equivalent
(SWE), as well as snow free water (SFW) representing the storage of melted snow. The water
stored in the stream channels and lakes is also included in the TWS estimate. Based on the



structure of PCR-GLOBWB, the total water storage (TWS) is computed as the sum of 27
different states: 2 groundwater, 8 soil moisture, 4 interceptions, 8 snow, 4 inundated top
water, and 1 surface water layer.
For each grid cell and for each daily time step, the model determines the water balance in two
vertically stacked soil layers and the groundwater store. The model also computes the vertical
water exchanges between the soil layers and between the inundated top water layer and the
atmosphere, i.e. rainfall and snowmelt, percolation and capillary rise, as well as evaporation
and transpiration fluxes. The active groundwater store underlies the soil, is fed by net
groundwater recharge, discharges to baseflow as a linear reservoir, and is exempt from the
direct influence of evaporation and transpiration fluxes. However, capillary rise from the
active groundwater store can occur depending on the simulated groundwater storage, the soil
moisture deficit, and the unsaturated hydraulic conductivity. Fluxes are simulated according
to the different land cover types. The model includes a physically-based scheme for
infiltration and runoff, resulting in the direct runoff, interflow, as well as groundwater
baseflow and recharge. River discharge is calculated by accumulating and routing the specific
runoff along the drainage network. For further details, including model parameterization, the
reader is referred to the technical reports and other relevant publications (van Beek and
Bierkens, 2009; van Beek, 2008; Sutanudjaja et al., 2011, 2014).

**4. Data and data processing**
**4.1 GRACE data**
The GRACE gravity product release 5 (RL05), generated by the University of Texas at
Austin's Center of Space Research (CSR, Bettadpur, 2012), was used as input. The product
consists of monthly sets of spherical harmonic coefficients (SHC) complete to the degree and
order 60.  On this basis, TWS variations were obtained for the study period between April
2002 and December 2010. The GRACE data were further processed in this study as follows:

- SHCs of degree 1 provided by Swenson et al. (2008) were restored, and all 5
  coefficients of degree 2 were replaced by the values estimated from satellite laser
  ranging (Cheng and Tapley, 2004).
- SHC variations were computed by removing the long-term mean (computed between
  April 2002 and December 2010) from each monthly solution.
- A destriping filter (Swenson and Wahr, 2006) was applied to the SHC variations. The
  filter used a $5^{th}$ degree polynomial (Savitsky-Golay) over a 5-point window to remove
  the correlations; orders below 8 remained unchanged.
- An additional 250-km radius Gaussian smoothing (Jekeli, 1981) was applied to SHC
  variations to suppress high-frequency noise, and the TWS variations ($\Delta\sigma$ [m]) were
  then computed using (Wahr et al 1998)

$$\Delta\sigma(\theta,\phi) = \sum_{l=1}^{60} \sum_{m=-l}^{l} W_l \overbrace{\frac{a_e(2l+1)}{3(1+k_l)}\frac{\rho_e}{\rho_w}}^{s_l} \Delta\bar{C}_{lm}\hat{Y}_{lm}(\theta,\phi),$$

(1)



where $\theta$, $\phi$ are co-latitude and longitude in spherical coordinates, $\Delta\bar{C}_{lm}$ is the SHC
variations of degree $l$ and order $m$, $\hat{Y}_{lm}$ is the normalized surface spherical harmonic,
$W_l$ is the Gaussian smoothing function, $S_l$ is a scaling factor used to convert
dimensionless coefficients to TWS in terms of Equivalent Water Heights (EWH), $a_e$
is the semi-major axis of the reference ellipsoid, $k_l$ is the load love number of degree
$l$, $\rho_e$ and $\rho_w$ are the average density of the Earth and water, respectively. In this study,
the TWS variations were computed at every 0.5°x0.5° grid cell. This cell size was
selected through trial and error as a balance between performance and resolution.
In general, filters suppress not only noise but also the genuine TWS signal, and are a well-
known source of signal leakage. To address this, a signal restoration method (Chen et al.,
2014; Tangdamrongsub et al., 2016) was employed. The method iteratively determined the
possible signal reduction caused by the filter applied and added it back to the filtered signals.
The errors of the procedure grew with the number of iterations, requiring a proper selection of
the convergence criterion. In this study, the criterion was chosen empirically: the signal
restoration process was iteratively repeated until the increment in every grid cell inside the
Hexi Corridor became smaller than 0.5 cm. This value is 2-3 times smaller than the GRACE
uncertainty (Wahr et al., 2006; Klees et al., 2008; Dahle et al., 2014). Figure 4 demonstrates
the signal restoration for October 2002. The convergence criterion was met after
approximately 6 iterations. The signal over the mountain range and Inner Mongolia became
apparent after the signal restoration was applied (see Fig. 4f).
**4.2 Forcing data**
The forcing data required by PCR-GLOBWB are precipitation, air temperature, and potential
evapotranspiration. Tangdamrongsub et al. (2015) showed that the use of high-quality
precipitation data may lead to better estimates of hydrological fluxes (e.g., TWS and
streamflow). In principle, local precipitation and surface temperature measurements could be
obtained from the China Daily Ground Climate Dataset provided by the China Meteorological
Data Sharing Service System (http://cdc.cma.gov.cn/home.do). A total of 23 weather stations
were found over the Hexi Corridor (see Fig. 1b). However, the measurements were spatially
sparse and did not cover the entire region. Therefore, the global precipitation data were used
to achieve a better spatial coverage. Four global precipitation products were considered for
inclusion:
•  The European Centre for Medium-range Weather Forecasts (ERA-Interim, spatial

277        resolution: 0.75°x0.75°; Dee et al., 2011)

•  The Tropical Rainfall Measuring Mission (TRMM 3B42, spatial resolution: 0.25°

279        x0.25°; Huffman et al., 2007; Kummerow et al., 1998)

•  The Climate Research Unit dataset (CRU, spatial resolution: 0.5° x0.5°; Mitchell and

281        Jones, 2005; van Beek, 2008)

•  The Princeton's Global Meteorological Forcing Dataset (Princeton, spatial resolution:

283        0.5° x0.5°; Sheffield et al., 2005)





To select the best product, the global precipitation values were firstly interpolated to the
weather station locations and then the correlation coefficient, Nash-Sutcliffe (NS) coefficient,
and RMS difference (RMSD) between the interpolated and observed ground data were
calculated. The mean values of the statistical estimates are shown in Fig. 5a. Overall, TRMM
provided the best data quality, with the highest correlation (~0.85) and NS coefficients
(~0.46), and an RMSD approximately 2–3 mm lower than other products. The high spatial
resolution of TRMM is probably the reason for its better performance. Therefore, this product
was chosen as the precipitation input. The low NS coefficient in all 4 cases suggests that the
coarse spatial resolution of the global precipitation datasets prevents them from capturing all
the local precipitation events.
A similar procedure was used to compare the air temperature data from ERA-Interim, CRU,
and Princeton. The statistical estimates are shown in Fig. 5b. Although the results from all
products were very similar, CRU provided the highest data quality in terms of correlation and
RMSD values, and therefore, it was used as the temperature input. As far as
evapotranspiration is concerned, few data are available for this region, so the data from (van
Beek, 2008) were used.
**4.3 Validation data**
**4.3.1 Groundwater**
Monthly groundwater well measurements at 5 locations (Fig. 1c) were obtained from the
ground network maintained by the Shiyang River Basin Management Bureau, and Institute of
Water Resources and Hydropower of Gansu Province. The in situ data were provided in the
form of piezometric heads, which needed to be converted to units of storage. For such a task,
several parameters, e.g., storage coefficient and specific yield are required, but they are not
available over the Hexi Corridor. To solve that problem, a scale factor computed using the
information from GRACE and the Global Land Data Assimilation System (GLDAS, Rodell et
al., 2004) was used for the conversion using the approach outlined by Tangdamrongsub et al.
(2015). In short, the procedure was as follows. First, GLDAS-based soil moisture storage
variations ($\Delta$SM) were removed from GRACE-derived TWS variations. Four variants of
GLDAS model (NOAH, CLM, MOSAIC, and VIC; see Rodell et al., 2004) were considered
and the average $\Delta$SM value was calculated. Taking into account that $\Delta$SM and groundwater
storage variations ($\Delta$GWS) are the major contributions to TWS variations, this resulted in
$\Delta$GWS ($\Delta$GWS$_{(GRACE-\Delta SM)}$). Then, by conducting a regression analysis between the monthly
time-series of piezometric head variation ($\Delta h$) and $\Delta$GWS$_{(GRACE-\Delta SM)}$, a scale factor ($f$) was
estimated using the following relationship:
$$\Delta\mathbf{GWS_{(GRACE-\Delta SM)}} + e = f \cdot \Delta h, \qquad\qquad (2)$$
where $e$ indicates the observation error. Finally, the in situ head measurements were
multiplied with the estimated scale factor ($\hat{f}$) to obtain the converted groundwater storage
variation ($\Delta$GWS$_{in\ situ}$) as:
$$\Delta\mathbf{GWS}_{in\ situ} = \hat{f} \cdot \Delta h. \qquad\qquad (3)$$






### 4.3.2 Streamflow

Monthly river gauge data were obtained from the same data centre as the groundwater
measurements. Due to the coarse spatial resolution of PCR-GLOBWB, it models only the
main river streams. Therefore, the gauge measurements of small river streams, as well as the
gauge measurements that contained many data gaps (e.g., more than 24 months), were
excluded. As a result, the measurements from only 2 gauges – at Xida and Xiying Rivers (see
Fig. 1c) – were used in this study.

### 4.4.3 Normalized Difference Vegetation Index (NDVI)

NDVI (Carlson and Ripley, 1997) is an indicator of vegetation health or "greenness". In this
study, NDVI and GWS were analysed to determine if the growing season was being extended
beyond the limited rainy period through groundwater extraction for irrigation. NDVI was
computed from the MODIS 8-day, 500-m spatial resolution surface reflectance product
(Vermote et al., 2011) based on data from Aqua satellite (MYD09A1 product). Based on the
location of the in situ groundwater measurements, the MODIS tiles h25v05 and h26v05 were
selected. First, the data were quality controlled: pixels with cloud cover were flagged and
filled values were masked. The 8-day NDVI was then computed as (Huete et al., 2002)
$$NDVI = \frac{\rho_{NIR} - \rho_R}{\rho_{NIR} + \rho_R},\qquad(4)$$
where $\rho_{NIR}$ and $\rho_R$ are the surface reflectance in the near-infrared and red portions of the
observed electromagnetic spectrum. The monthly-averaged NDVI was then computed based
on the derived 8-day NDVI values. Note that NDVI values range between –1 and 1, and a
value over 0.2 generally indicates vegetation.

### 5. Methodology and implementation

### 5.1 Ensemble Kalman Filter (EnKF)

The Ensemble Kalman Filter (EnKF; Evensen, 2003) is used to assimilate GRACE derived
TWS into the PCR-GLOBWB model. The EnKF works in two steps, a forecast step and
analysis (update) step. The forecast step involves propagating the states forward in time using
the model (PCR-GLOBWB). Identical to how the EnKF is implemented by Forman et al.
(2012), the state vector ($\boldsymbol{\psi}(t)$ in this study is an $nm$ x 1 vector, where $n = 27$ is the number of
TWS-related states from PCR-GLOBWB (see Sect. 3), and $m$ is the number of model grid
cells. The model estimates are related to the GRACE observations by
$$\boldsymbol{d}(t) = H\boldsymbol{\psi}(t) + \epsilon;\ \epsilon \sim \mathcal{N}(0, R),\qquad(5)$$
where $\boldsymbol{d}(t)$ is an $m$ x 1 vector containing the GRACE observations for the month of interest,
and $H$ is a measurement operator which relates the PCR-GLOBWB state $\boldsymbol{\psi}(t)$ to the
observation vector $\boldsymbol{d}(t)$. Notice that the number of observations is equal to the number of grid





cells because the GRACE-based estimates are obtained for all the grid cells of the PCR-
GLOBWB model (see Sect. 4.1). The uncertainties in the observations are given in the
random error $\epsilon$, which is assumed to have zero mean and covariance matrix $\mathbf{R}_{m \times m}$. As the
sum of all state elements at a given cell is equal to TWS, the $\mathbf{H}$ matrix is defined as:
$$\mathbf{H} = \left[ \begin{pmatrix} (1\,1\,1\dots 1)_{1 \times n} & 0 & \cdots & 0 \\ 0 & (1\,1\,1\dots 1)_{1 \times n} & \cdots & 0 \\ \vdots & \vdots & \ddots & \vdots \\ 0 & 0 & \cdots & (1\,1\,1\dots 1)_{1 \times n} \end{pmatrix} \right]_{m \times nm}. \tag{6}$$

If the ensembles of the states are stored in a matrix
$\mathbf{A}_{nm \times N} = \big( \psi_1(t), \psi_2(t), \psi_3(t), \dots, \psi_N(t) \big)$, the ensemble perturbation matrix is defined as
$\mathbf{A}' = \mathbf{A} - \overline{\mathbf{A}}$, where $\overline{\mathbf{A}}$ is the mean computed from all ensemble members. Similarly, the
members of the GRACE observation vector are stored in the matrix
$\mathbf{D}_{m \times N} = \big( d_1(t), d_2(t), d_3(t), \dots, d_N(t) \big)$. Notice that $\mathbf{D}_{m \times N}$ contains $N$ identical columns.
The analysis equation can be expressed as (Evensen, 2003)
$$\mathbf{A}^a = \mathbf{A} + \Delta\mathbf{A} = \mathbf{A} + \mathbf{K}(\mathbf{D} - \mathbf{H}\mathbf{A}) \tag{7}$$
with
$$\mathbf{K} = \mathbf{P}_e\mathbf{H}^T(\mathbf{H}\mathbf{P}_e\mathbf{H}^T + \mathbf{R})^{-1}, \tag{8}$$
where $\mathbf{A}^a_{nm \times N}$ is the updated model state, $\Delta\mathbf{A}_{nm \times N}$ is the update from Kalman filter, and
$\mathbf{K}_{nm \times m}$ is the Kalman gain matrix. The model error covariance matrix $(\mathbf{P}_e)_{nm \times nm}$ is
computed as
$$\mathbf{P}_e = \mathbf{A}'(\mathbf{A}')^T/(N-1). \tag{9}$$
The computation of matrix $\mathbf{R}$, the error variance-covariance matrix of GRACE data in the
spatial domain, is discussed in Sect. 5.2.2.
In the initialization phase, to obtain the initial states, the model was spun up between 1
January 2000 and 31 December 2000 as a hot start. This time interval was sufficient to reach
the dynamic equilibrium. The obtained initial state $\psi(t)$ was perturbed and $N = 100$
ensemble members $\psi_i(t), i = 1, 2, 3, \dots, N$ were generated. The $N = 100$ ensemble runs
between 1 January 2001 and 31 March 2002 were then conducted independently based on the
perturbed initial states. This resulted in ensemble spread of the estimated states. The model
was then propagated in time between 1 April 2002 and 31 December 2010 without
assimilating any observation. This case is referred to hereafter as the Ensemble Open Loop
(EnOL). For the EnKF, the model was also propagated beginning from 1 April 2002, but the
observations (when available) were assimilated.
The processing diagram is shown in Fig. 6, and follows the methodology introduced by
Forman et al. (2012). The state is first propagated in time from the first to the last day of the
month without applying DA, and the monthly averaged states are calculated from the daily
values. When the GRACE observation for that month is available, the DA routine is activated.



Otherwise, the model continues propagating to the next month without applying DA. The DA
routine updates the TWS-related states using Eq. (7). The daily increment (DINC) of the
update is then computed by dividing the monthly averaged update by the total numbers of
days in that month (numday$_{month}$). The model propagation is then restarted (second run), using
the last day of the previous month (month-1, numday$_{month-1}$) as the initial state. In this second
run, the DINC is added to the initial states every day up to the last day of the month. The DA
scheme is repeated to the end of the study period.
Spatial correlations of model and observation errors were also taken into account. De Lannoy
et al. (2009) proposed a so-called 3D-Fm (3-dimentional fine scale with multiple observation)
approach, which is called EnKF 3D in this paper. The approach only considered the spatial
correlations between the neighbouring grid cells. This reduced the computational cost, as only
a small subset of cells pairs was considered instead of all cells pairs. That approach was
applied not only to observation errors, but also to model errors in TWS and TWS-related
components in this study. The EnKF 3D scheme is illustrated in Fig. 7. For a particular grid
cell (centre grid cell), all TWS-related components of the neighbouring grid cells and the
centre grid cell are used to form the state ($\mathbf{A}^s_{np \times N}$) and observation ($\mathbf{D}^s_{p \times N}$) matrices, where $p$
is the number of the considered grid cells. The matrix notation with superscript $\mathbf{s}$ (e.g., $\mathbf{A}^s$) is
only used to emphasize the cell-dependent version, and it can be substituted into the original
matrix notation (e.g., $\mathbf{A}$) in Eqs. (5–9). It is emphasized here that EnKF 3D involves only $p$
grid cells instead of all $m$ grid cells. As such, the measurement operator, model error
covariance matrix, and observation error covariance matrix becomes $\mathbf{H}^s_{p \times np}$, $(\mathbf{P}^s_e)_{np \times np}$, and
$\mathbf{R}^s_{p \times p}$, respectively. In this study, the neighbouring grid cells were assumed to be the ones
inside the Gaussian smoothing radius applied, i.e., 250 km. The EnKF was then applied and
the states of the centre grid cell (only) were updated. The procedure was repeated through all
grid cells. Note here that to avoid the edge effects, the grid cells at the edge were not updated
in the computation. To investigate the impact of including spatial correlations of errors, the
EnKF 1D was also considered. The EnKF 1D scheme is similar to EnKF 3D, but the spatial
correlations are omitted (i.e., the off-diagonal elements of the covariance matrices $\mathbf{P}^s_e$ and $\mathbf{R}^s$
are set to zero).
Furthermore, sampling errors caused by finite ensemble size might lead to spurious
correlations in the estimated model error covariance matrices (Hamill et al., 2001). To reduce
such an effect, a distance-dependent localization function is applied to $\mathbf{P}^s_e$ (pair-wise). In this
study, the Gaussian function ($c(\alpha)$) (Jekeli, 1981) was used:
$$c(\alpha_{j_1,j_2}) = \frac{e^{-b[1-\cos(\alpha_{j_1,j_2}/a_e)]}}{1-e^{-2b}} \qquad (10)$$
with $b = \frac{\ln(2)}{1-\cos(L/a_e)}$, $\qquad (11)$
where $\alpha_{j_1,j_2}$ is the distance on the Earth surface between two grid cells ($j_1$ and $j_2$), and $L$ is the
correlation distance. The variogram analysis was used to derive the TWS correlation distance
($L$) of PCR-GLOBWB, assuming that it is similar to the correlation distance of model errors.
It was found to be approximately equal to 110 km over the Hexi Corridor. For GRACE





observations, to ensure that the spurious error correlations at distances greater than the
Gaussian smoothing distance, 250 km, was insignificant, the localization applied to $\mathbf{R}^s$ was
based on $L = 250$ km.

**5.2 Errors of PCR-GLOBWB model and errors in GRACE observations**
**5.2.1 Model errors**
The two primary sources of considered errors in the PCR-GLOBWB model are the
meteorological forcing data and the model parameters. For forcing data, the precipitation
uncertainties were quantified as the RMS error provided by the TRMM product (Huffman,
1997). The uncertainties of temperature and potential evapotranspiration were not provided as
parts of the corresponding products, and therefore errors of $2^o$C, and 30% of the nominal
potential evapotranspiration value were assumed, respectively. The error levels were chosen
through trial-and-error, mainly to allow the ensemble to grow between updates. The
precipitation and potential evapotranspiration were perturbed with additive lognormal noise
while the temperature was perturbed with additive Gaussian noise. The forcing data
uncertainties were assumed to be spatially correlated, which was accounted for using an
exponential decay function. Based on a variogram analysis, the correlation distances of
precipitation, temperature and potential evapotranspiration were found to be approximately
150 km, 450 km, and 450 km, respectively.
As far as model parameters are concerned, a total of 15 TWS-related parameters (see Table 1)
were perturbed using additive Gaussian noise without spatial correlations. The standard
deviation of the perturbations of the parameters was set to 20% of the range of the nominal
values.
**5.2.2 GRACE observation errors**
Spatial correlations of GRACE observation errors were also taken into account in the DA
scheme. The uncertainties in the GRACE-derived TWS over the Hexi Corridor were
computed using the monthly calibrated error variance-covariance matrix of the SHCs ($\mathbf{\Sigma}$)
provided by the CSR. Recalling the replacement of the low degree SHCs (see Sect. 4.1), the
error (co-)variances of SHCs degree 2 were not provided by Cheng and Tapley (2004), and
therefore the values obtained from the CSR were used. As for SHCs of degree 1, the error (co-
) variances were not available from (Swenson et al., 2008) either and were set to zero. Note
that $\mathbf{\Sigma}$ only reflects the error of the original GRACE data, i.e. before the GRACE processing
described in Sect. 4.1 was applied. To obtain the error variance-covariance matrix associated
with the post-processed GRACE data, an ensemble of SHC noise realizations $\mathbf{Q}^c$ was first
generated based on $\mathbf{\Sigma}$ as follows:
$$\mathbf{Q}^c = (\mathbf{\Sigma})^{\frac{1}{2}} \mathbf{Q}^w, \tag{12}$$
where $\mathbf{Q}^w = (q_1^w, q_2^w, q_3^w, \dots, q_N^w)$ contains a set of white noise realizations and has the
dimension of $s \times N$, where $s = 1891$ is the number of SHCs, and $N = 100$ is the number of



realizations. The matrix $\mathbf{Q}^c = (q_1^c, q_2^c, q_3^c, ..., q_N^c)$ has the same dimension as $\mathbf{Q}^w$ and contains
an ensemble of correlated noise realizations in SHCs. Then, each noise realization (i.e.,
column of $\mathbf{Q}^c$) was post-processed in the same way as the GRACE data (Sect. 4.1), which
resulted in $\widehat{\mathbf{Q}}^c = (\hat{q}_1^c, \hat{q}_2^c, \hat{q}_3^c, ..., \hat{q}_N^c)$. The post-processing included applying the destriping and
Gaussian smoothing filters, as well as the signal restoration using the same number of
iterations as was used in the GRACE data post-processing. The error variance-covariance
matrix $\widehat{\mathbf{\Sigma}}$ associated with the SHCs after post-processing was then computed as
$$\widehat{\mathbf{\Sigma}} = \left[\widehat{\mathbf{Q}}^c \left(\widehat{\mathbf{Q}}^c\right)^T\right] / (N - 1). \tag{13}$$
Recalling Eq. (1), the TWS variations over the Hexi Corridor can be computed as
$$\Delta\mathbf{\sigma} = \mathbf{YSx}, \tag{14}$$
where $\Delta\mathbf{\sigma}$ is the vector composed of the computed TWS variations at grid cells, $\mathbf{Y}$ is the
matrix of spherical harmonic synthesis (cf. Eq. (1)), $\mathbf{S}$ is the matrix containing the scaling
factors $S_l$, and $\mathbf{x}$ is the vector composed of the dimensionless SHC variations after GRACE
data post-processing described in Sect. 4.1. Then, the error covariance matrix $\mathbf{R}$ of the
GRACE-based TWS variations over the Hexi Corridor was computed with the error
propagation law as
$$\mathbf{R} = \mathbf{YS}\,\widehat{\mathbf{\Sigma}}\,(\mathbf{YS})^T. \tag{15}$$
Some statistics of GRACE TWS errors over the Hexi Corridor are shown in Fig. 8. The error
standard deviation in Oct. 2002 varied with location (Fig. 8a), whereas the error correlation
showed a distance-decay pattern in all directions (Fig. 8b). The areally-averaged standard
deviations over 4 basins stayed in most of the months at a similar level of approximately 1 cm
(Fig. 8c). The large uncertainty in September 2004 was likely caused by the near-repeat orbit
of GRACE satellites during that month.

**6. Results and discussion**
The structure of this section is as follows. First, the impact of assimilation using EnKF 3D on
the total TWS is considered in Sect. 6.1. Then, the impact of the EnKF 3D on the estimates of
the individual stores is investigated in Sect. 6.2. The performances of the EnKF 1D and EnKF
3D schemes are compared in Sect. 6.3 in terms of total TWS and the individual stores.
Finally, in Sect. 6.4 the assimilation results are used together with ancillary remote sensing
data to study water resources in the Hexi Corridor.

**6.1 Performance of GRACE DA scheme**
To demonstrate the impact of DA, Fig. 9 shows the daily TWS estimates over the Shiyang
River Basin between 1 April 2002 and 31 December 2003. Several features associated with





the EnKF can be observed. Firstly, when a GRACE observation is available, the EnKF moves
the estimated TWS towards it. As a result, the estimated TWS lies between the EnOL estimate
and the GRACE observation most of the time. Secondly, the standard deviation across the
EnKF ensemble of TWS values is smaller than that of the EnOL and smaller than the GRACE
observation error. Thirdly, at the first month (April 2002) the TWS estimates of the EnOL and
EnKF were similar at the forecast step (as the initial states were the same), but became
different when the daily increment was applied to the EnKF (see point (a) in Fig. 9). Finally,
discontinuities in the time-series before the update were observed at the end of the month e.g.,
in November and December 2002 (point (b) and (c)), and February 2003 (point (d)).
Applying the daily increment (see Sect. 5.3) served as a smoother, and these stepwise changes
were reduced. Similar features were also seen in the EnKF 1D TWS estimates (not shown).

**6.2 Impact of GRACE DA on individual stores**
The monthly-averaged values of the TWS and individual stores in each of the 4 basins are
presented in Fig. 10. Overall, TWS estimates over the Hexi Corridor mostly reflect variations
of SM and GWS components, while snow and surface water are minor contributors,
constituting less than 5% in most basins. Clear seasonal variations in TWS were seen in all
basins for GRACE, EnOL and GRACE DA (both EnKF 1D and EnKF 3D) (Fig. 10 a,b,c,d).
As observed in Fig. 9, the GRACE DA estimated TWSs are generally between the GRACE
observations and the EnOL estimates. As a result of assimilating GRACE data, both the EnKF
1D and EnKF 3D added water to all basins between 2002 and 2005 and reduced it from the
basins between 2006 and 2010. This is also reflected in the time-series of SM (Fig. 12 e,f,g,h )
and GWS (Fig. 12 I,j,k,l). Additionally, the annual amplitudes and phases of GRACE DA
estimated TWS were also found mostly in between the values computed from the GRACE
observations and the EnOL results (see Table 2). In particular, the GRACE-DA estimated
TWS's phase was always closer to the GRACE observation. The phase shifts of
approximately 1 month were seen in both GRACE DA estimated TWS and GRACE
observations compared to the EnOL results. Similar phase differences of approximately 1
month were also observed in SM and GWS components.
Differences in the long-term trends were also detected between the TWS estimates from the
model alone (EnOL) and the GRACE DA. The GRACE DA results showed similar
decreasing TWS trends to the GRACE data, while the EnOL showed increasing trends (Fig.
10 a,b,c,d, see also Table 7). This change in TWS trend was clearly a result of assimilating
GRACE observations. The negative trends were also observed after DA in the GWS
component in most basins (Fig. 10 i,j,l). This indicates the potential of GRACE DA in
adjusting GWS. In this way, one can reveal continued groundwater consumption to support
local agricultural activities (Li et al., 2013). Unlike over other basins, the negative trend of
GWS estimates was not clearly present over the Desert Region (Fig.10k). This could be due to
the small size of the groundwater store of this region (see also below), and most of the update
took place in the SM component. As a result, a relatively large negative trend was seen in SM





rather than GWS after GRACE DA (see also Table 7). Further discussions of the trends are
given in Sect. 6.4.
The impact of GRACE DA on different stores was influenced by both the model parameters
and the forcing data assigned. For example, comparing to other basins, the Shiyang River
Basin has smaller values of saturated hydraulic conductivities ($K_{sat,up}$, $K_{sat,low}$) and a higher
value of the recession coefficient (J) (see Table 3), which allows for larger updates to the
groundwater compartment. This explains the significant amount of update seen in GWS (Fig.
10i). Similar behaviour was also reported in (Tangdamrongsub et al., 2015). Conversely, the
groundwater compartment of the Desert Region received a tiny amount of water due to the
high value of $K_{sat}$ and low value of J.
Forcing data also had an impact on which stores were updated. Consider, for instance, the SM
compartment. Although the Shiyang River Basin has the smallest $K_{sat}$ compared to other
basins (Table 3), the annual amplitude of SM (~ 1 cm) there was not the smallest, but the
largest one (see Fig. 10e), as the basin received the greatest amount of rainfall (see Fig. 2a). In
contrast to the Shiyang River Basin, the Shule River Basin has similar $K_{sat}$ but the smallest
SM annual amplitude (~0.3 cm) was observed (Fig. 10h). This reflects the limited amount of
rainfall received by the basin (see Fig. 2g).
Snow estimates (SWE plus SFW) were very small (less than 0.2 cm) over the Hexi Corridor
and therefore were only slightly updated by GRACE DA. The large amount of snow seen as
the sharp peaks, e.g., in January 2008 was caused by the precipitation and temperature
variability. In January 2008, the precipitation records were 159 % higher than the January
average value while the temperature was $2 – 3^{o}C$ lower. Such a condition resulted in a large
amount of snow. Finally, GRACE DA influences the surface water, but the amplitude is still
lower than that of the GRACE uncertainties. Validation of the surface water estimates in
terms of river streamflow is given in Sect. 6.3.2.

**6.3 Impact of taking spatial correlations of errors into account**
The impact of taking spatial correlation of errors into account was evaluated by comparing
estimates from EnKF-1D and EnKF-3D with the in situ measurements and independent
satellite observations. First of all, the impact of accounting for the error correlations was
clearly seen in the TWS estimates (Fig. 10 a,b,c,d). When the error correlations were ignored
(EnKF 1D), the TWS estimate received a larger update from GRACE, particularly between
2002 and 2005. Hence, the estimate was drawn significantly closer to the observation.
Presence of error correlations effectively reduces amount of information in GRACE data,
since spatial averaging of such data mitigates noise to a much less extent than averaging of
data with uncorrelated errors. Therefore, the impact of GRACE data in the EnKF 3D case is
reduced. As such, the EnKF 3D estimated TWS was always between the EnOL and EnKF 1D
results. Validating against the in situ groundwater and streamflow data will reveal which
results are closer to the truth (Sect. 6.3.1, 6.3.2).



Taking error correlations into account also has a clear impact on the SM and GWS
components. For SM, similarly to TWS, the EnKF 1D yielded a larger update between 2002
and 2005 compared to the EnKF 3D (Fig. 10 e,f,g,h). The difference between EnKF 1D and
3D results became smaller after 2005. This can be attributed to the fact that the ensemble
spread in the SM component becomes smaller after several years of updates. After 2005, the
ensemble spread of SM was lower than the GRACE uncertainty, and therefore taking the error
correlations into account did not have a significant impact on the SM estimates. For GWS, the
impact of taking error correlations into account was even clearer, especially in terms of the
long-term trend (Fig. 10, i,j,k,l). With the exception of the Desert Region, the EnKF 1D
showed a steeper decreasing trend in all basins. For snow and surface water, the impact of
considering error correlations was not significant due to the fact that the stores are small, as
compared to SM and GWS.
**6.3.1 Validation against groundwater estimates**
The GWSs estimated from GRACE DA were validated against the well measurements at 5
locations shown in Fig. 1c. The GWS estimate at each well location is shown in Fig. 11.
Compared to the EnOL results, GRACE DA results were visually closer to the well
measurements at all 5 locations. The EnKF 1D and EnKF 3D showed a noticeable difference
at each location. The updated GWS estimates were evaluated in terms of the correlation
coefficient, RMS difference (RMSD), and long-term trend (Table 4, 5). Overall, the EnOL
resulted in relatively poor correlation coefficients at most stations (except station W1), with
the average value of only 0.06. Clear improvements were seen after GRACE DA was applied.
The average correlation coefficient increased to approximately 0.6 – 0.7. Although the EnKF
1D introduced a greater update than the EnKF 3D, it only showed higher correlation
coefficients at stations W1 and W3. Applying the EnKF 3D led to correlation coefficients
greater than 0.45 in all stations, and on average it improved the correlation coefficient by
approximately 0.1 over EnKF 1D. In terms of RMSD, applying GRACE DA reduced the
difference by approximately 15 – 25% compared to the EnOL. Compared to EnKF 1D, the
EnKF 3D significantly improved the RMSD in most stations. The EnKF 1D only performed
better than EnKF 3D at station W1, where it reduced the RMSD by approximately 16 %
compared to 8% reduction by the EnKF 3D. The noticeably low GWS observed by the well
data at station W2 in the summers of 2007 and 2008 (Fig. 11b) was probably caused by
significant groundwater abstraction. These local features could not be reproduced due to a
limited spatial resolution of the model and GRACE observations. As a result, neither of the
EnKF algorithms could improve the GWS estimates at the W2 location during those periods.
The long-term trend estimated between 2007 and 2010 was also used to evaluate the impact of
taking the error correlations into account (Table 5). The EnOL trend estimates were
considered poor as they showed the largest RMS difference respected to the in situ data. In
fact, they were the least consistent with the in situ estimates at each individual station. Similar
to the results in terms of correlation coefficient and RMSD (see Table 4), the EnKF 3D led to
the largest improvement in the trend estimates (RMSD=0.54 compared to 0.93 after EnKF
1D). However, while the EnKF 3D showed closer long-term trends to the in situ





measurements at stations W2, W4, W5, the EnKF 1D produced better estimates at station W1
and W3.
Thus, both EnKF 1D and 3D led to the improvement of the GWS estimates in terms of all
metrics. In terms of the average results and at the majority of well locations, the EnKF 3D
provided more improvement than the EnKF 1D.

### 6.3.2 Validation against streamflow estimates

The streamflow estimates were validated against the river gauge measurements at locations
G1 and G2 (Fig 1c). Results are shown in Figure 12 and Table 6. Only modest improvements
in the streamflow estimates are observed in terms of the correlation coefficient, NS
coefficient, and RMS difference (RMSD). This behaviour is similar to what was observed
previously for Rhine River Basin when a different hydrology model and input data were used
(Tangdamrongsub et al., 2015). Figure 12 shows that taking error correlations into account
had little impact, i.e. similar streamflow estimates were seen for EnKF 1D and 3D results. At
location G1 (Fig. 12a), GRACE DA added more water to the stream channel between 2002
and 2006 and reduced it between 2008 and 2010. This behaviour is consistent with the TWS
estimates discussed in Sect. 6.2. GRACE DA increased the correlation coefficient from 0.82
to 0.84, increased the NS coefficient from 0.65 to 0.69, and reduced the RMSD by
approximately 5 % (Table 6). Similar improvements were also observed at G2.
Comparing to the gauge measurements, both the EnOL and GRACE DA overestimated the
streamflow in September 2007 and September 2008 at G2. The sudden surge in streamflow
results from heavy rainfall while the soil is saturated (Fig. 13). For example, in September
2007, the second highest amount of SM storage in the record (~19.5 cm) was observed when
the third largest amount of rainfall (~3 mm) occurred. Similarly, in September 2008, large SM
storage (~20 cm) and the heaviest rainfall (~3.4 mm) forced PCR-GLOBWB to generate a
large amount of streamflow, which significantly exceeded the actual one observed at G2.
Inaccurate precipitation data and model calibration led to the discrepancy here. GRACE DA
was unable to reduce these spurious peaks due to the limited spatial (~250 km) and temporal
(1 month) resolution.

### 6.4 Declining water storages in the Hexi Corridor

The water resources situation over the Hexi Corridor was assessed using long-term trends
estimated from 9-year EnKF 3D results. This DA variant is primarily discussed here as it
provided better agreement with in-situ observations than the EnKF 1D (see Sect. 6.3.1). For
completeness, however, the values estimated from GRACE, EnOL, EnKF 1D, and
precipitation are also provided. The trends in the TWS, SM and GWS variations for the 4
basins, as well as the areally-averaged values for the entire Hexi Corridor, are given in Table
7. The average EnKF 3D trends are all negative: approximately –0.2, –0.1, and –0.1 cm/yr for
TWS, SM. and GWS, respectively. This reduction in the water storages is observed despite
the increased amount of rainfall, which shows a positive trend of about 0.4 (mm/month)/yr.



The water storage reductions can likely be attributed to the extraction of groundwater to meet
irrigation demands. In Sect. 6.5, it will be shown that groundwater extractions are essential for
that purpose in the Hexi Corridor.
Focusing on individual river basins provides additional insight into the water storage issue, as
the influence of the large desert area is removed. The water storage losses in the individual
basins are even more pronounced, particularly in the Shiyang River Basin. This basin had the
greatest TWS loss (approximately 0.4 cm/yr), which was entirely caused by the reduction of
GWS. This can be explained by groundwater abstraction to meet the irrigation demand in the
region. The Heihe and Shule River Basins also experienced a TWS loss of ~0.2 cm/yr, which
came from a reduction of both SM and GWS storages. Again, the negative GWS trend was
likely caused by significant pumping of groundwater to maintain crop production. This is
consistent with the extreme water stress over the Heihe River basin between 2001 and 2010,
which was documented in Table 11.7 of the study by Chen et al. (2014). Finally, the
decreasing TWS trend of 0.1 cm/yr detected in the Desert Region was dominated by loss of
SM storage.

## 6.5 Connection to agriculture activity

Figure 14 shows the monthly averaged groundwater head measurements at wells W1 to W5 in
the Shiyang River Basin (Fig. 1c). Monthly averaged precipitation and NDVI values are
shown as well. Since extracted water can be used to support agriculture not only at the well
location but also in the nearby area, precipitation and NDVI are reported as the average values
within a circular area of the 10-km radius. These data will be used to ascertain if groundwater
extractions to support agriculture might be the source of the negative GWS trends observed in
Fig. 11 and Table 6. From Fig. 14, it is noticed that the growing period is approximately
between May and October, where the amount of rainfall is higher than 15 mm/month and the
NDVI is typically greater than 0.2. By observing well measurements, precipitation, and NDVI
together, some groundwater extraction signatures can be explained by the extension of the
growing period over the dry season. For example, at station W1, the groundwater in 2010 was
lower than the average, showing a gradual decrease in summer (Fig. 14a). One may attribute
this to the shortage of rainfall in July and August 2010, which was lower than the average
(Fig. 14b). However, the NDVI value was higher than the average during summer 2010 (Fig.
14c), which implies that water from other sources than precipitation was probably used to
maintain the growing period. This additional water was likely extracted from the ground, and
such an activity led to a decreased groundwater table during summer 2010. A similar
explanation can be applied to station W2, where low groundwater head, low rainfall, and high
NDVI were observed in summer 2007 and summer 2008 (Fig. 14 d,e,f). At station W3, the
behaviour is similar to station W1: the extension of the growing period was observed in
summer 2010, where the GWS and precipitation were lower than the average, while NDVI
was significantly higher (Fig. 14 g,h,i). Groundwater pumping signatures were not present at
stations W4 and W5.




## 7. Conclusions

This study improved the estimation of water resources dynamics in the Hexi Corridor by assimilating GRACE-derived TWS variations into the PCR-GLOBWB hydrological model. It was found that including the spatially-correlated errors into the DA scheme led to the improvement of the state estimates. Furthermore, GRACE DA estimates revealed the reduction of water storages between 2002 and 2010. The Shiyang River Basin suffered the most from the water loss, which was likely caused by the overuse of the groundwater for irrigation. Due to inaccurate groundwater abstraction information, PCR-GLOBWB alone could not properly capture the downward trend of water storages. This highlights the value of the GRACE DA in this situation.

Furthermore, we demonstrate how the error covariance matrix of GRACE TWS can be derived from the error covariance matrix of GRACE SHCs (which is currently provided together with the SHCs themselves). This study shows that it is necessary to consider the error correlations in the DA scheme. To illustrate, the assimilation schemes considered 2 variants of the error variance-covariance matrix, excluding and including error correlations. Validating against ground data found that both DA schemes lead to noticeable improvement in the state estimates in terms of correlation, RMS difference, and long-term trend. However, ignoring error correlations in DA tended to over-fit results to the observations, and in many cases led to less accurate state estimates. On average, greater improvement was seen when the error correlations were taken into account. At the same time, we admit that the derivation of GRACE TWS error variance-covariance matrices is very computationally demanding. Still, we believe that this is a reasonable price to pay as full (and only full) error covariance matrices reflect the real uncertainty of GRACE observations. Of course, the performance of EnKF 3D needs to be further investigated in other geographical locations and with different hydrological models to confirm the benefits of this scheme.

GRACE DA strengthened the PCR-GLOBWB model in capturing the signature of groundwater abstraction. It should be emphasized that GRACE does not fix a technical problem of the hydrological model, but it rather provides information which is not available otherwise. Note that, in principle, the model may predict any long-term behaviour of water storage, but that information should be brought in "by hand" (e.g., via the groundwater abstraction parameter). As soon as that information is not available, reliable long-term predictions on the basis of hydrological modelling alone are conceptually impossible. GRACE DA acts as a provider of a missing puzzle piece here.

Substantial decreasing of water storage was observed in the Hexi Corridor. The region received increased precipitation between 2002 and 2010, but the water storage was found to be declining, particularly over the Shiyang River Basin. The amount of water from rainfall was obviously insufficient to support irrigation water requirements. Irrigation water demands increased significantly to maintain the crop production and, as a result, the region was under extreme water stress. Water consumption from all available sources was essential for bridging the deficit, including an enormous amount of groundwater extraction. This study illustrates



the connection between groundwater pumping and agriculture activity, which is clearly seen
from the ground observation and remote sensing data.
To further improve the DA performance, an extended or an alternative DA framework can be
considered. One of the points of attention is only a minor improvement in streamflow
estimates, which is caused by an insufficient temporal and spatial resolution of GRACE data.
A promising way to go is to improve the runoff scheme at a conceptual level, e.g., by
extending GRACE DA with a simultaneous parameter calibration. To that end, the state
vector should be extended to include also selected model parameters (Eicker et al., 2014;
Wanders et al., 2014). This allows for the adjustment of the storage size and might lead to a
more accurate estimate of model states, including streamflow (Wanders et al., 2014).
Alternative ensemble-based DA approaches, such as particle filters (Weerts and El Serafy,
2006), can also be considered. Particle filters estimate a sample from the true posteriori
distribution, which is not necessarily Gaussian, like in the EnKF. The approach has been
shown very effective for the parameter calibration (Dong et al., 2015). Finally, the uses of
improved gravity solutions from the GRACE Follow-on (Flechtner et al., 2014) will probably
further increase the accuracy of the GRACE DA estimates.

**Acknowledgement**
This research was funded by the Nederlandse Organisatie voor Wetenschappelijk Onderzoek
(Netherlands Organisation for Scientific Research, NWO; project number 842.00.006) and
Ministry of Science and Technology of China (MoST, Project Number 2010DFA21750)
under the Samenwerking China - Joint Scientific Thematic Research Programme (JSTP). The
research was also sponsored by the NWO Exacte Wetenschappen, EW (NWO Physical
Sciences Division) for the use of supercomputer facilities, with financial support from NWO.
The research was also co-funded by National Natural Science Foundation of China (NSFC,
project number 51279076).

**Appendix A:** Water demand calculation in PCR-GLOBWB
PCR-GLOBWB includes an online and integrated scheme to simulate irrigation water
requirement. The scheme separately parameterizes two different irrigated crop groups: paddy
and non-paddy, aggregated from 26 crop classes from the MIRCA2000 dataset, which
accounts for various growing season lengths. The calculated irrigation water requirement is
applied according to crop specific calendars, which ensure optimal crop growth. Principally,
this irrigation water demand scheme aims to maintain certain soil moisture levels in order to
provide optimal crop transpiration but still takes into account soil water availability,
interception, bare soil evaporation, as well as open water evaporation over inundated paddy
fields. Over daily time steps, irrigation water demand is calculated by considering the deficit
of readily available water in the soil moisture layers (thickness <= 1.2 m) to their total storage
capacities. To represent flooding irrigation over paddy fields, a certain surface water depth





(e.g., 50–100 mm) is maintained until the late crop development stage before the harvest. The
dynamic irrigation scheme in PCR-GLOBWB also considers the historical growth of irrigated
areas based on FAOSTAT.
Other sectoral water demands, including those from livestock, industry, and household, are
compiled from several sources by considering many factors, including past change in
population, socio-economic and technological development.
• Livestock water demand is calculated by multiplying the number of livestock in a grid
cell with its corresponding drinking water requirement, which is a function of air
temperature. The gridded global livestock densities of cattle, buffalo, sheep, goats,
pigs and poultry and their corresponding drinking water requirements are obtained
from FAO (Steinfeld et al., 2006), and FAOSTAT (http://faostat.fao.org/).
• Historical and gridded industrial demand data are obtained from several sources
(Shiklomanov, 1997; WRI, 1998; Vörösmarty et al., 2005). The algorithm to calculate
this demand includes country-specific economic development based on four
socioeconomic variables: gross domestic product (GDP), electricity production,
energy consumption, and household consumption. Associated technological
development per country is then approximated by energy consumption per unit
electricity production, which accounts for industrial restructuring or improved water
use efficiency.
• Household or domestic water demand is estimated by multiplying the number of
population in a cell with the country-specific per capita domestic withdrawals. The
country domestic withdrawals are taken from the FAO AQUASTAT
(http://www.fao.org/nr/water/aquastat/main/index.stm) and (Gleick et al., 2009).
Economic and technological developments are taken into account. Seasonality of
household/domestic water demand is also considered, using air temperature as a
proxy. Here, available gridded global population maps per decade (Klein Goldewijk
and van Drecht, 2006) are used to downscale the country-scale map to produce the
gridded water demand data.
The allocation of various water sources, i.e. surface water, groundwater and desalinated water,
is principally based on the simulation of their availabilities. However, the model also
incorporates some inventory data and studies about local preferences of certain regions. This
means that areas with extensive surface water irrigation supplies and networks prioritize
surface water use. Moreover, cities with poor drinking water facilities and water distribution
networks use groundwater as their main source (although they may be close to the rivers with
abundant surface water).

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












**Table 1**. PCR-GLOBWB model parameters related to the TWS estimate. Parameters are
functions of spatial coordinates, except DDF which is a constant.

| Parameter | Description | unit |
|---|---|---|
| $K_{sat,up}$ | Saturated hydraulic conductivity of the upper soil storage | m/day |
| $K_{sat,low}$ | Saturated hydraulic conductivity of the lower soil storage | m/day |
| $SC_{up}$ | Storage capacity of the upper soil | m |
| $SC_{low}$ | Storage capacity of the lower soil | m |
| $f_g^{min}, f_f^{min}, f_p^{min}, f_{np}^{min}$ | Minimum soil depth fraction of grassland (g), forest (f), paddy irrigation (p), non-paddy irrigation (np) | - |
| $f_g^{max}, f_f^{max}, f_p^{max}, f_{np}^{max}$ | Maximum soil depth fraction of grassland (g), forest (f), paddy irrigation (p), non-paddy irrigation (np) | - |
| J | Groundwater recession coefficient | 1/day |
| DDF | Degree-day factor in the snow pack | $^{o}$Cm/day |
| $KC^{min}$ | Minimum crop coefficient | - |


**Table 2**. TWS, SM and GWS estimated annual amplitude (A, cm) and phase (P, month) in 4
different basins computed between April 2002 and December 2010. Areally averaged values
for the entire Hexi Corridor are also given.

| | | | Shiyang | Heihe | Desert | Shule | Areally-average |
|---|---|---|---|---|---|---|---|
| TWS | GRACE | A | $2.05 \pm 0.31$ | $1.49 \pm 0.21$ | $1.79 \pm 0.23$ | $1.21 \pm 0.27$ | $1.43 \pm 0.18$ |
| | | P | $6.97 \pm 0.29$ | $6.80 \pm 0.27$ | $6.49 \pm 0.24$ | $8.61 \pm 0.42$ | $7.05 \pm 0.24$ |
| | EnOL | A | $1.35 \pm 0.16$ | $0.90 \pm 0.07$ | $0.66 \pm 0.07$ | $0.37 \pm 0.06$ | $0.70 \pm 0.06$ |
| | | P | $6.35 \pm 0.23$ | $5.61 \pm 0.14$ | $5.80 \pm 0.19$ | $5.40 \pm 0.31$ | $5.74 \pm 0.16$ |
| | EnKF 1D | A | $1.61 \pm 0.16$ | $0.87 \pm 0.10$ | $1.05 \pm 0.11$ | $0.40 \pm 0.11$ | $0.80 \pm 0.09$ |
| | | P | $6.96 \pm 0.19$ | $6.80 \pm 0.22$ | $6.47 \pm 0.19$ | $8.35 \pm 0.51$ | $6.92 \pm 0.23$ |
| | EnKF 3D | A | $1.49 \pm 0.13$ | $0.80 \pm 0.08$ | $0.72 \pm 0.07$ | $0.26 \pm 0.09$ | $0.72 \pm 0.07$ |
| | | P | $6.42 \pm 0.17$ | $6.12 \pm 0.19$ | $6.40 \pm 0.20$ | $8.48 \pm 1.02$ | $6.44 \pm 0.22$ |
| SM | EnOL | A | $1.03 \pm 0.11$ | $0.70 \pm 0.06$ | $0.62 \pm 0.07$ | $0.31 \pm 0.05$ | $0.59 \pm 0.06$ |
| | | P | $5.77 \pm 0.20$ | $5.60 \pm 0.16$ | $5.82 \pm 0.21$ | $5.03 \pm 0.32$ | $5.62 \pm 0.18$ |
| | EnKF 1D | A | $0.88 \pm 0.09$ | $0.75 \pm 0.09$ | $0.99 \pm 0.11$ | $0.36 \pm 0.10$ | $0.67 \pm 0.08$ |
| | | P | $6.55 \pm 0.21$ | $7.01 \pm 0.22$ | $7.08 \pm 0.21$ | $8.47 \pm 0.54$ | $7.26 \pm 0.24$ |
| | EnKF 3D | A | $1.30 \pm 0.10$ | $0.66 \pm 0.07$ | $0.71 \pm 0.08$ | $0.12 \pm 0.08$ | $0.55 \pm 0.07$ |
| | | P | $5.59 \pm 0.15$ | $6.25 \pm 0.20$ | $6.44 \pm 0.20$ | $8.19 \pm 0.37$ | $6.32 \pm 0.22$ |
| GWS | EnOL | A | $0.50 \pm 0.08$ | $0.19 \pm 0.03$ | $0.02 \pm 0.004$ | $0.09 \pm 0.01$ | $0.12 \pm 0.01$ |
| | | P | $7.84 \pm 0.29$ | $7.13 \pm 0.26$ | $5.43 \pm 0.34$ | $6.91 \pm 0.29$ | $7.22 \pm 0.21$ |
| | EnKF 1D | A | $0.65 \pm 0.05$ | $0.12 \pm 0.03$ | $0.01 \pm 0.01$ | $0.05 \pm 0.01$ | $0.10 \pm 0.01$ |
| | | P | $8.69 \pm 0.16$ | $7.82 \pm 0.40$ | $7.91 \pm 1.90$ | $8.49 \pm 0.29$ | $8.32 \pm 0.25$ |





| | | | | | | |
|---|---|---|---|---|---|---|
| EnKF 3D | A | 0.70 ± 0.06 | 0.11 ± 0.02 | 0.02 ± 0.01 | 0.05 ± 0.01 | 0.10 ± 0.01 |
| | P | 8.52 ± 0.16 | 7.50 ± 0.31 | 7.76 ± 1.00 | 8.66 ± 1.33 | 8.26 ± 0.23 |


**Table 3**. Averaged values and standard deviations of precipitation and model parameters for 4
different basins.

| | Shiyang | Heihe | Desert | Shule |
|---|---|---|---|---|
| Precipitation (mm/month) | 21 ± 12 | 13 ± 12 | 11 ± 2 | 8 ± 6 |
| $K_{sat,up}$ (m/day) | 0.42 ± 0.24 | 0.71 ± 0.69 | 1.16 ± 0.89 | 0.42 ± 0.15 |
| $K_{sat,low}$ (m/day) | 0.24 ± 0.15 | 0.61 ± 0.50 | 0.93 ± 0.74 | 0.24 ± 0.05 |
| J (1/day) | 0.0057 ± 0.0088 | 0.0024 ± 0.0049 | 0.0018 ± 0.0017 | 0.0013 ± 0.0017 |



**Table 4**. Statistical values of the GWS computed from the in situ well measurement and
GRACE DA estimates between January 2007 and December 2010. The average values are
computed by averaging the estimated statistical values from all well locations.

| | | W1 | W2 | W3 | W4 | W5 | Average value |
|---|---|---|---|---|---|---|---|
| Correlation coefficient [-] | EnOL | 0.74 | 0.17 | -0.04 | -0.05 | -0.53 | 0.06 |
| | EnKF 1D | 0.84 | 0.32 | 0.90 | 0.45 | 0.64 | 0.63 |
| | EnKF 3D | 0.82 | 0.49 | 0.85 | 0.51 | 0.83 | 0.70 |
| RMS difference [cm] | EnOL | 0.69 | 1.67 | 0.77 | 3.34 | 3.81 | 2.06 |
| | EnKF 1D | 0.58 | 1.63 | 0.40 | 2.56 | 2.58 | 1.55 |
| | EnKF 3D | 0.63 | 1.43 | 0.38 | 2.24 | 1.27 | 1.19 |


**Table 5.** Long-term trends and standard deviations of the in situ data and the DA estimates.
The RMS difference (RMSD) between the in situ data and the DA trend estimates are also
provided.

| | **W1** | **W2** | **W3** | **W4** | **W5** | **RMSD** |
|---|---|---|---|---|---|---|
| In situ | -0.49 ± 0.03 | 0.01 ± 0.06 | -0.60 ± 0.004 | 0.56 ± 0.12 | -1.40 ± 0.03 | 0 |
| EnOL | -0.57 ± 0.01 | -0.64 ± 0.002 | -0.01 ± 0.01 | -1.69 ± 0.01 | 1.29 ± 0.02 | 1.62 |
| EnKF 1D | -0.52 ± 0.02 | -0.58 ± 0.04 | -0.74 ± 0.02 | -1.33 ± 0.08 | -1.99 ± 0.13 | 0.93 |
| EnKF 3D | -0.83 ± 0.02 | -0.51 ± 0.03 | -0.38 ± 0.01 | -0.44 ± 0.08 | -1.18 ± 0.06 | 0.54 |


**Table 6**. Statistical values of the streamflow computed from the river stream gauge
measurement and GRACE DA estimates between April 2002 and December 2010. The
average values are calculated by averaging the estimated statistical values from both gauge
locations.

| | | G1 | G2 | Average value |
|---|---|---|---|---|
| Correlation coefficient [-] | EnOL | 0.82 | 0.76 | 0.79 |
| | EnKF 1D | 0.84 | 0.77 | 0.81 |
| | EnKF 3D | 0.84 | 0.78 | 0.81 |





| NS coefficient [-] | EnOL | 0.65 | 0.56 | 0.61 |
|---|---|---|---|---|
| | EnKF 1D | 0.69 | 0.57 | 0.63 |
| | EnKF 3D | 0.69 | 0.57 | 0.63 |
| RMS difference [cm] | EnOL | 5.49 | 3.09 | 4.29 |
| | EnKF 1D | 5.18 | 3.08 | 4.14 |
| | EnKF 3D | 5.23 | 3.04 | 4.14 |



**Table 7**. TWS, SM, GWS, and precipitation estimated long-term trends in 4 different basins
computed between April 2002 and December 2010. Areally averaged values for the entire
Hexi Corridor are also given.

| | | Shiyang | Heihe | Desert | Shule | Areally-average |
|---|---|---|---|---|---|---|
| TWS (cm/yr) | GRACE | -0.73 ± 0.04 | -0.64 ± 0.03 | -0.72 ± 0.03 | -0.34 ± 0.04 | -0.59 ± 0.03 |
| | EnOL | 0.30 ± 0.02 | 0.24 ± 0.01 | 0.20 ± 0.01 | 0.18 ± 0.01 | 0.22 ± 0.01 |
| | EnKF 1D | -0.72 ± 0.02 | -0.41 ± 0.01 | -0.33 ± 0.02 | -0.34 ± 0.02 | -0.39 ± 0.01 |
| | EnKF 3D | -0.36 ± 0.02 | -0.21 ± 0.01 | -0.11 ± 0.01 | -0.25 ± 0.01 | -0.20 ± 0.01 |
| SM (cm/yr) | EnOL | 0.38 ± 0.02 | 0.21 ± 0.01 | 0.17 ± 0.01 | 0.14 ± 0.01 | 0.19 ± 0.01 |
| | EnKF 1D | -0.11 ± 0.01 | -0.20 ± 0.01 | -0.29 ± 0.02 | -0.22 ± 0.01 | -0.23 ± 0.01 |
| | EnKF 3D | 0.10 ± 0.01 | -0.12 ± 0.01 | -0.12 ± 0.01 | -0.14 ± 0.01 | -0.11 ± 0.01 |
| GWS (cm/yr) | EnOL | -0.08 ± 0.01 | 0.03 ± 0.004 | 0.02 ± 0.001 | 0.04 ± 0.002 | 0.02 ± 0.002 |
| | EnKF 1D | -0.61 ± 0.01 | -0.16 ± 0.004 | -0.01 ± 0.002 | -0.12 ± 0.001 | -0.16 ± 0.002 |
| | EnKF 3D | -0.39 ± 0.01 | -0.09 ± 0.003 | 0.01 ± 0.001 | -0.11 ± 0.001 | -0.11 ± 0.002 |
| **Precipitation** ((cm/month)/yr) | | 0.04 ± 0.01 | 0.04 ± 0.01 | 0.05 ± 0.01 | 0.02 ± 0.01 | 0.04 ± 0.01 |





**Figure 1**. Geography of the Hexi Corridor. (a) Land cover and division into individual regions
(Shiyang River Basin, Heihe River Basin, Shule River Basin, and a Desert), (b) Topography
and locations of the local meteorological stations (triangles), (c) Zoom-in on the Shiyang
River Basin, showing the locations of considered groundwater wells and river stream gauges.






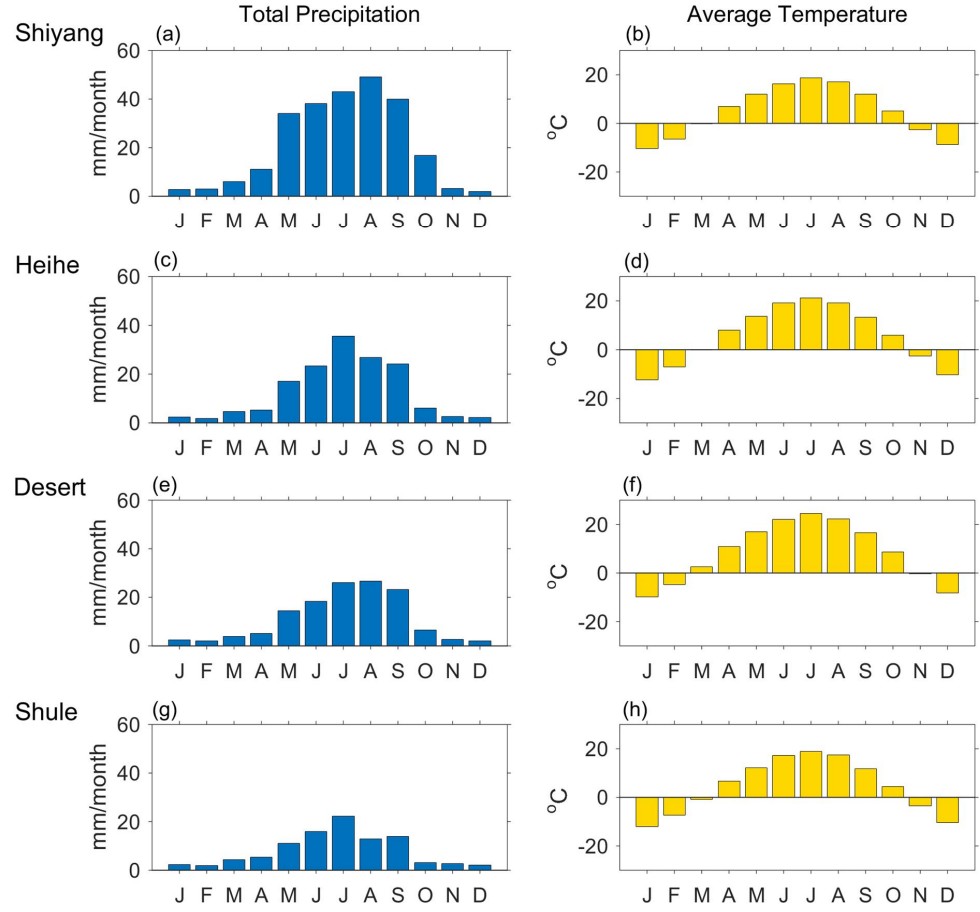

**Figure 2**. Monthly total precipitation and averaged temperature over 4 regions of the Hexi
Corridor.




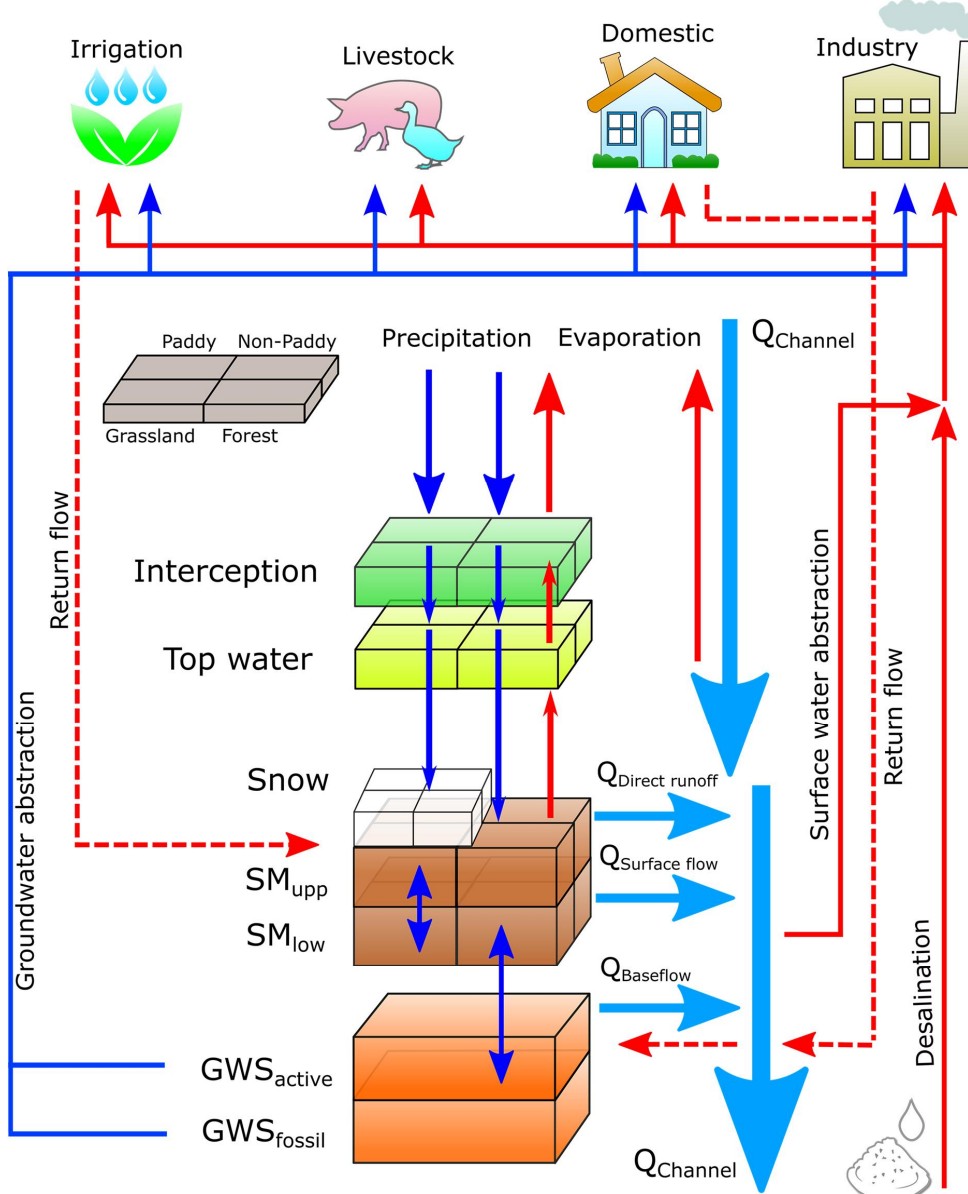


**Figure 3**. The structure of PCR-GLOBWB hydrological model.





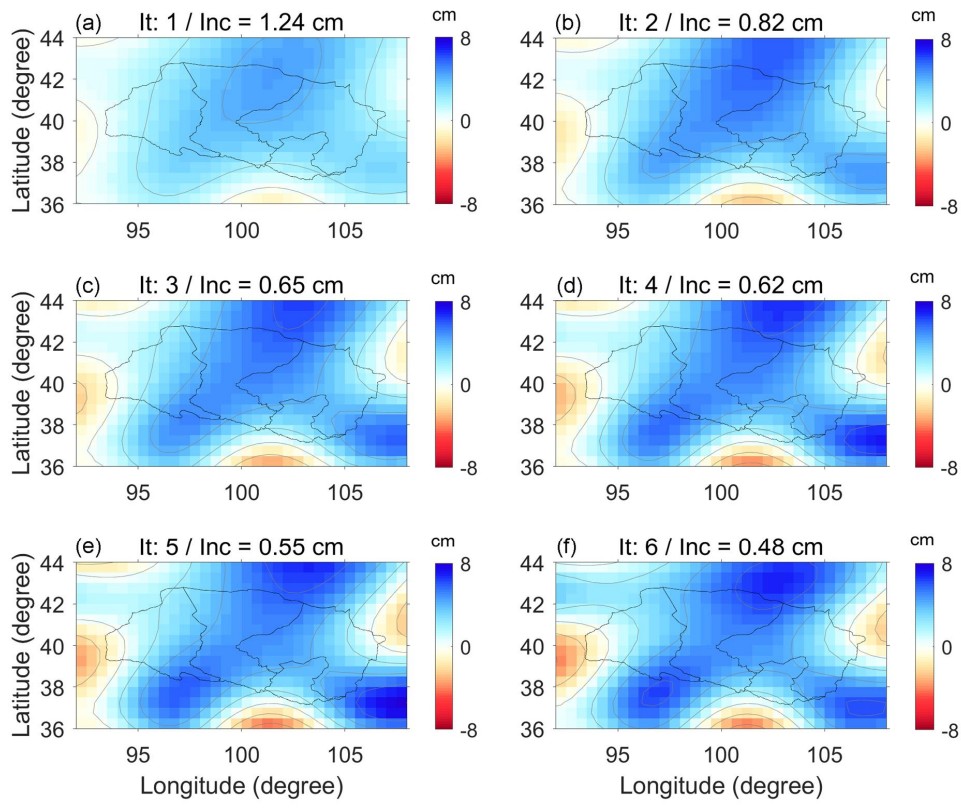


**Figure 4**. GRACE-derived TWS variation of October 2002. The signal restoration was
applied to restore the signal mitigated by the applied spatial filter. After each iteration (It), the
increment (Inc) was computed. The procedure was stopped after 6 iterations when the
increment was lower than 0.5 cm (f).








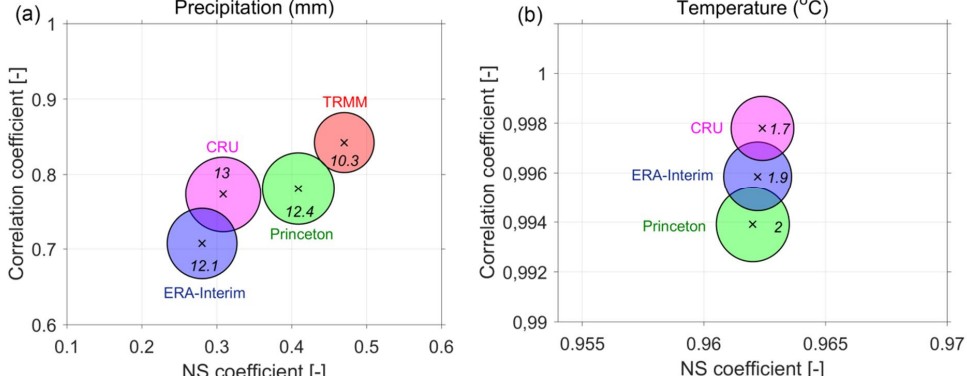


**Figure 5**. The correlation coefficient, NS coefficient, and RMS difference computed between
the local and different global forcing data. The RMS difference is shown as the radius of the
circle (also explicitly provided as the number).







Figure 6. DA diagram representing the disaggregation of monthly averaged TWS from GRACE into the daily PCR-GLOBWB state estimates.





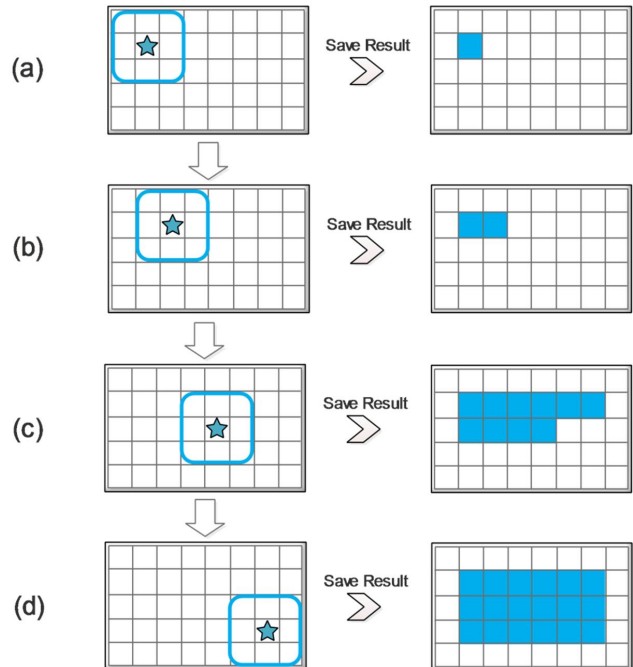

1100

**Figure 7**. Demonstration of EnKF 3D scheme, accounting for the spatially-correlated errors.
For a centre grid cell, the state and observation matrices contain all TWS-related components
of the neighbouring grid cells and the centre grid cell (left). The covariance matrices $\mathbf{P_e}$ and $\mathbf{R}$
are computed based on the data from these grid cells. Then, the EnKF is applied and the states
of the centre grid cell are updated (right). The procedure is repeated through all grid cells.
















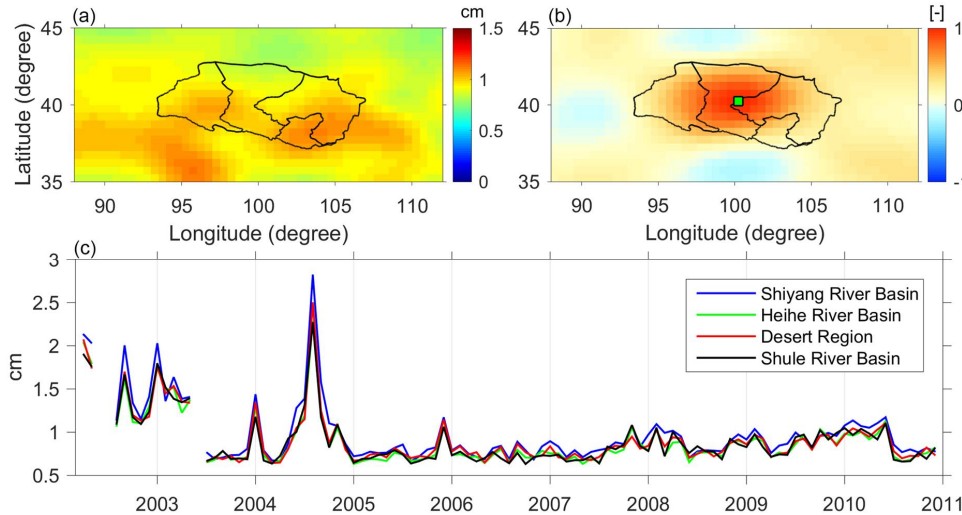

**Figure 8.** Some statistics of errors in GRACE-derived TWS variation over the Hexi Corridor.
The standard deviation (a) and the correlation coefficient with respect to the green point (b)
for a sample month, October 2002, are shown in the top. The time-series of averaged standard
deviation computed over 4 different basins are shown in the bottom plot (c).




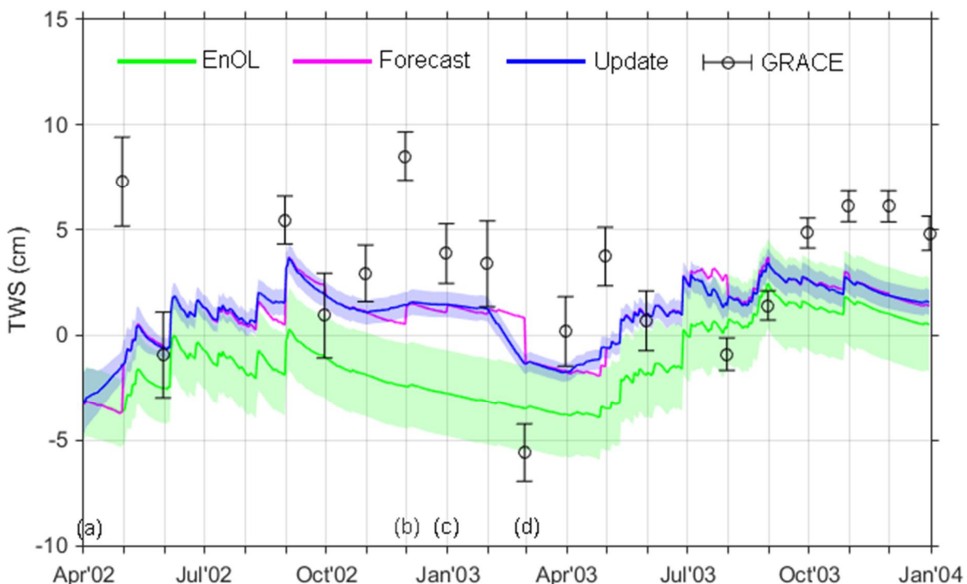

1127

**Figure 9**. Daily TWS variations estimated between 1 April 2002 and 31 December 2003,
averaged over Shiyang River Basin. The mean value of the ensemble is given as the solid line,
and the standard deviation is shown as the shaded envelope. The TWS estimates from model
only (EnOL), GRACE DA forecast (EnKF before the update), GRACE DA update (EnKF
after update), and GRACE observations are shown. The x-axis labels represent the first day of
the month. Some features of the DA scheme regarding the identical TWS estimate seen at the
beginning of the update (point a) and the observed spurious jumps (point b,c,d) are also
shown.






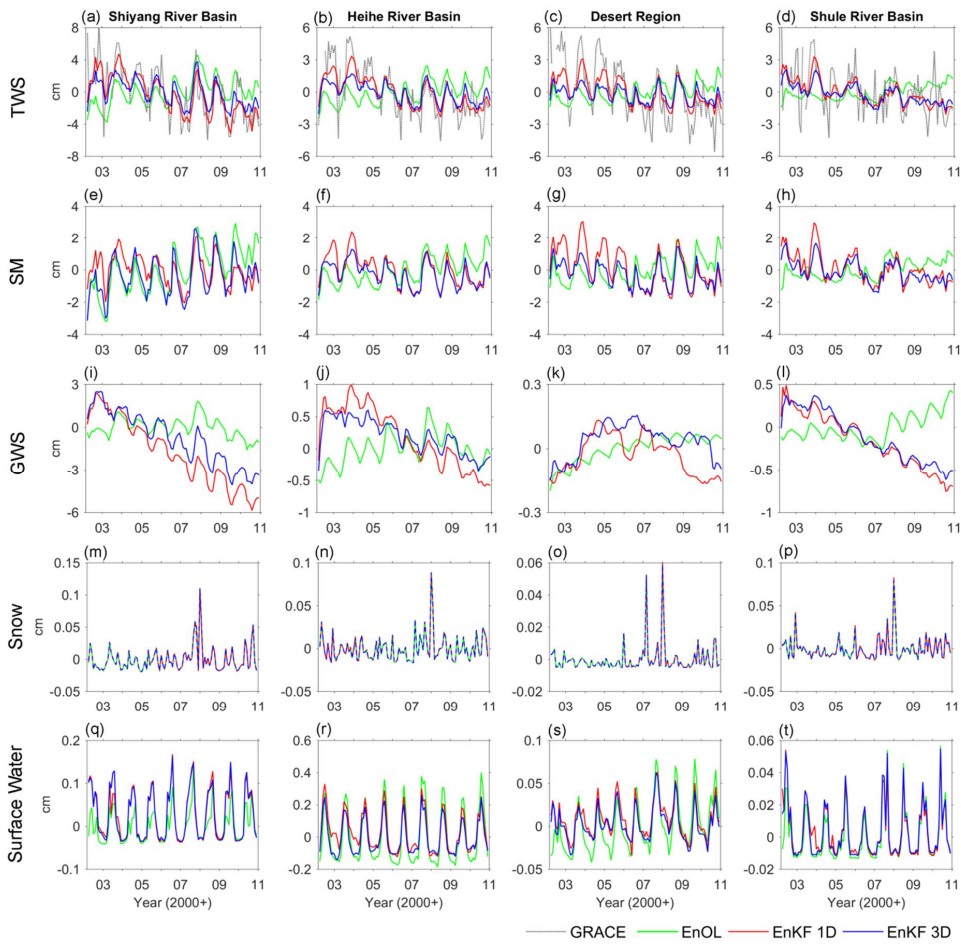


**Figure 10**. Monthly TWS, soil moisture (SM), groundwater storage (GWS), snow, and
surface water variation estimated between April 2002 and December 2010 from the EnOL,
EnKF 1D, EnKF 3D, and GRACE observations over 4 basins.





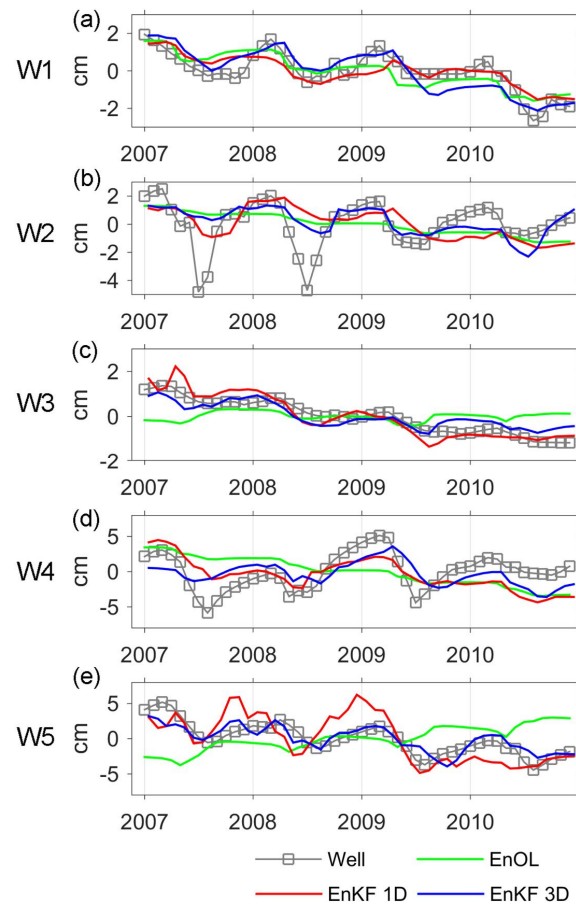


**Figure 11**. Monthly GWS variation estimates from the in situ well measurements, as well as
EnOL, EnKF 1D, and EnKF 3D results, between January 2007 and December 2010 at 5
groundwater well locations. The chosen period is based on the availability of the in situ data.





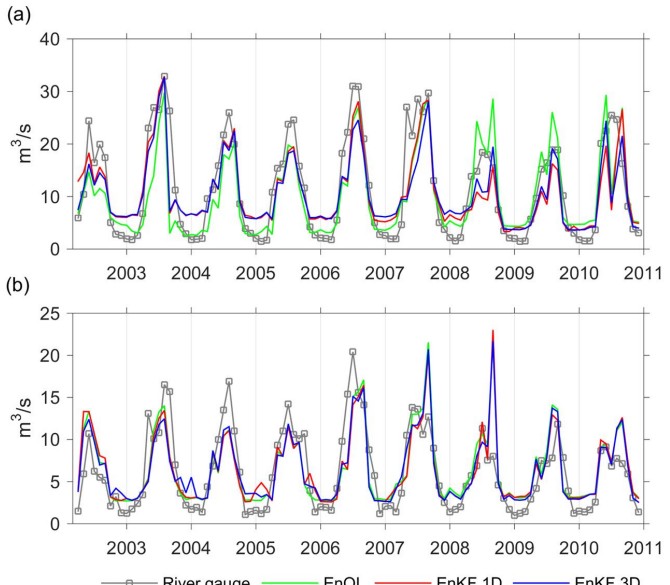


**Figure 12**. Monthly streamflow estimates from the in situ river gauge measurements, as well
as EnOL, EnKF 1D, and EnKF 3D results, between April 2002 and December 2010 at 2 river
gauge locations, G1 (a) and G2 (b).

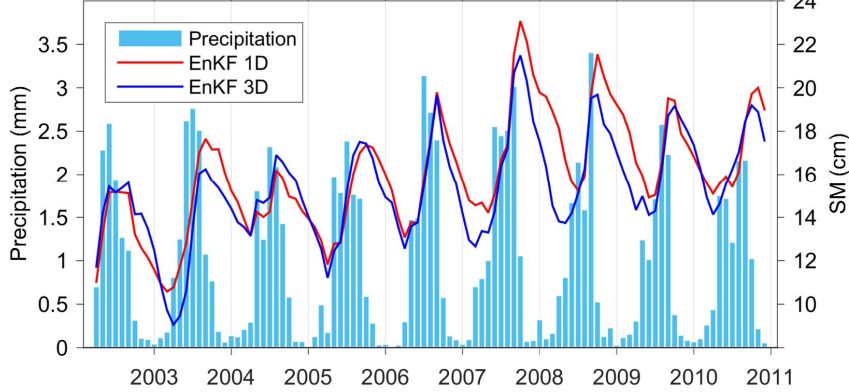


**Figure 13**. Monthly total precipitation (mm) and SM estimates (cm) from EnKF 1D and
EnKF 3D results at river gauge G2 location.






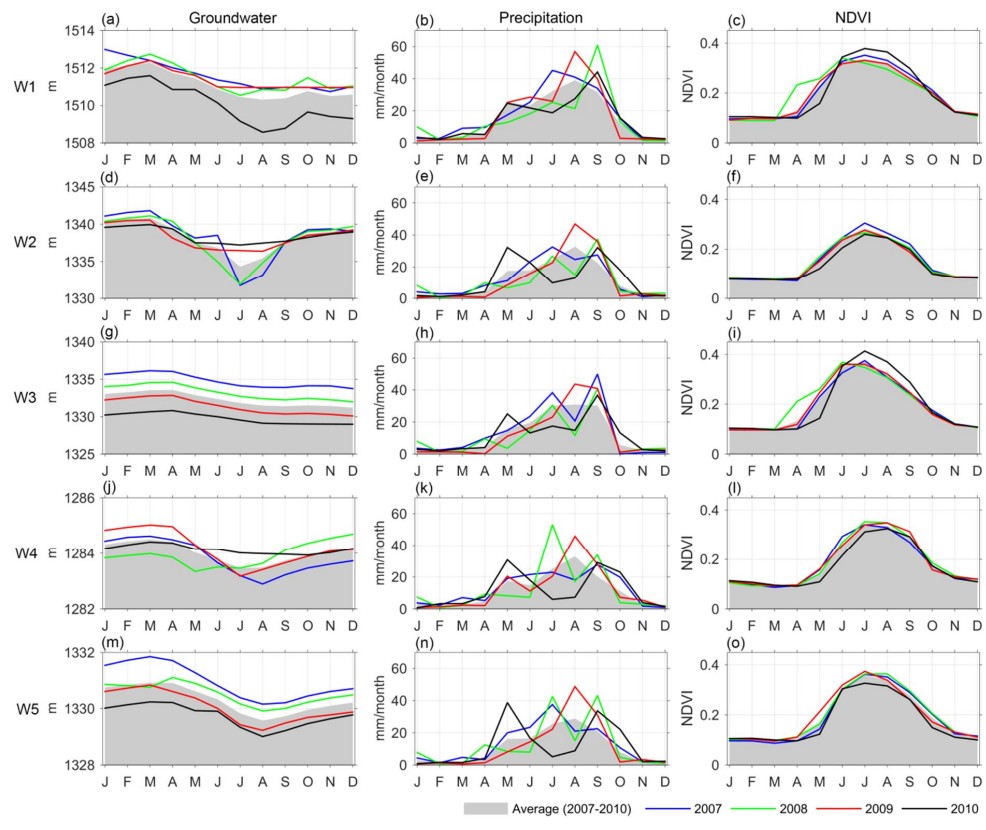


**Figure 14**. The monthly averaged groundwater head measurement (left), total precipitation (middle) and NDVI (right) for 5 groundwater well locations. Precipitation and NDVI data are reported as the average values within the circular areas of the 10-km radius. The long-term average values between January 2007 and December 2010 are shown in the grey shed, and the values in 2007, 2008, 2009, and 2010 are shown as blue, green, red, and black lines, respectively. The period is chosen based on the availability of the well data.

1164