# Peer review of "Improving estimates of water resources in a semi-arid region by assimilating GRACE"

_Hydrology and Earth System Sciences, 2016_

## Referee Comment (RC1) · H. Bogena (Referee) · 19 Sep 2016

This MS describes a study about the assimilation of water storage estimates derived from the GRACE satellite mission data into the hydrological model PCR-GLOBWB using an Ensemble Kalman Filter approach for the Hexi Corridor in Northern China. The authors found that area-averaged values of TWS, soil moisture, and groundwater storages over the region decreased with an average rate of approximately 0.2, 0.1, and 0.1 cm/yr in terms of equivalent water heights, respectively. They concluded that this decrease was likely due to the groundwater consumption required to maintain the growing period.

The MS is very well written and presents an interesting GRACE DA model application

for the analysis of the effects of groundwater consumption on water storage in an arid region. Thus, it fits well to the scope of this journal. However, there are some issues regarding the structure, the methods and interpretations of the results (see comments below).

General comments concerning the hydrological modelling:

The interactive modules for simulating water abstraction etc. with the PCR-GLOBWB model are described in greater detail, but these did not be used in this study. Thus the model description should focus on the considered process.

The model parameterisation with respect to the soil hydraulic properties needs to be better described.

I suggest adding some plots showing the special distribution of simulated TWS for the different DA scenarios.

In the DA scheme only TWS is considered. It is no clear, how "added" or "subtracted" water was distributed by DA to the different model storages (e.g. SM, GW, snow).

Compared to the model results the variations in GRACE determined TWS are much more pronounced. Possible reasons should be discussed in greater (e.g. influence of the pattern restauration procedure).

It is unclear if at all or how groundwater abstraction was considered in the modelling. If this was considered, why was the groundwater abstraction not considered in the DA (e.g. by updating the groundwater abstraction parameter)?

Specific comments

Title: The term "semi-arid" is not correct (see below)

At times TWS variations are simply termed "TWS". This is somewhat confusing. The terms "TWS variations" or short "TWSV" should be always used.

[Figure]

L45: The groundwater well data should integrate of smaller areas than the catchment area of the streamflow data. Therefore, I am not convinced that this is a problem of spatial resolution.

L57-59: According to the Köppen climate classification this region belongs to "cold desert climate" (BWk).

L67-68: This depends largely on the measured variable. For instance, streamwater discharge data provides integrated information for large catchment areas.

L81: In addition, hydrological models typically suffer from inadequate process representations (model structure errors).

L98: "jump" of what?

L115: What is the size of the area?

L115-118: How do you know (e.g. the watershed area of the Rhine river is much smaller than the Hexi Corridor area)? Can you provide the SNR values for these different areas?

L122: What is the difference between "surface water" and "inundated water"?

L128-129: In which way are the results validated against remote sensing?

L147: The term "basin" is not appropriate.

L181: "distributed hydrological model"

L184-185: Also indicate the temporal resolution of the model.

L185-193: It is unclear, how or if at all these interactive modules for simulating water abstraction etc. have been used in this study. Clearly it was not the focus of this study. Thus I suggest removing this section incl. Appendix A.

L197: Delete "an"

L208: Change "states" into "water storages"

L219: This is rather a conceptual model.

L230: Explain "complete to the degree and order 60"

L259: Does this increment correspond to the monthly change in TWS?

L261: Is this the general uncertainty of GRACE?

L263-264: By looking at Fig. 4 this procedure seems to have mainly intensified the already existing pattern. To which extent are the temporal variations in TWS estimates influenced by this procedure?

L287-289: It is well-know that global precipitation products show considerable uncertainties, which is also indicated by the low NS values. Since in-situ data is available, I suggest to correct the TRMM data product using the approach suggested by Condom et al. (2011).

L298: Actual or potential ET?

L327-329: Actually, more appropriate data is available from other gauging stations in the Hexi Corridor for this study (see e.g. Zhang et al., 2015, 2016).

L307-322: Because of this conversion method any comparison of groundwater storage changes from in-situ and GRACE observations will not be independent. This needs to be discussed in some detail. In addition, in the procedure described in Tangdamrong-sub et al. (2015) two parameter were used instead of one. Please comment on this difference.

L317-318: Please provide a figure with the data and the regression.

L320: Why are you using an averaged f value to calculate the groundwater storage for each well? I would have thought that the variations in parameter f should represent local variations in storage parameters of the aquifers. Please explain the reasoning

behind this procedure.

L451: Please explain how you selected these parameters (e.g. did you use a sensitivity test?).

L526-527: Change into Figure 10. . .

L545: "on" instead of "of"

L549-550: Please provide information on the origin of these parameter values.

L543: How do you know that the groundwater store of the Desert Region is small.

L553-554: Please explain in greater detail, why higher values of K_sat and lower values of J have led to a smaller amount of water addition.

L599-600: I wonder whether the better agreement with the GRACE DA results is due to (or a least partly du to) the scaling procedure of the piezometer data. Please add a discussion on this.

L642: Clearly, predictions for G2 were improved to a lesser degree.

L647-648: These are very low amounts of precipitation, indicating very local precipitation events. It would be interesting to see the spatial distribution of these rainfall events and the resulting modelled soil moisture distribution.

L676-678: Why should the SM storage of the Desert Region decrease although precipitation shows an increasing trend? Please discuss.

L712-714: Until now, there was no indication that groundwater abstraction was considered in the modelling. Please add a description. Why was the groundwater abstraction not considered in the DA?

L734-735: See comment above. Would it be possible to update the groundwater abstraction parameter?

L744: Please provide quantitative information on groundwater abstraction.

Literature

Condom T, Rau P, Espinoza JC (2011) Correction of TRMM 3B43 monthly precipitation data over the mountainous areas of Peru during the period 1998–2007. Hydrol Process 25:1924–1933. doi:10.1002/hyp.7949

Zhang L, Nan Z, Xu Y, Li S (2016) Hydrological Impacts of Land Use Change and Climate Variability in the Headwater Region of the Heihe River Basin, Northwest China. PLoS ONE 11(6): e0158394. doi:10.1371/journal.pone.0158394.

Zhang A, Zheng C, Wang S, Yao Y. (2015): Analysis of streamflow variations in the Heihe River Basin, northwest China: trends, abrupt changes, driving factors and ecological influences. Journal of Hydrology: Regional Studies. 3:106–24. doi: 10.1016/j.ejrh.2014.10.005.

---

## Referee Comment (RC2) · Anonymous Referee #2 · 23 Sep 2016

Tangdamrongsub et al. show the improvement of a regional hydrologic model using GRACE DA. Using GRACE data the authors can show the decline of water storages. The study further shows the importance of including error correlations in the DA scheme. The authors successfully show the added value of a GRACE DA including error correlations and therefore provide a valuable tool for regional groundwater estimation and data scarce region. However, I do have some open issues regarding GRACE processing models, the available groundwater data and geology, as well as the error estimation. Overall, I congratulate the authors for a well written and structured paper and suggest the paper for publication after addressing the below mentioned comments.

Different GRACE gravity field models are available, CSR (this study, p6, l227ff), GFZ,

[Figure]

JPL, CNES/GRGS (Sakumura et al 2014). Why was CSR selected and how are the differences between the different GRACE processing models for the study region. I understand that the focus of the article is on the added valued of the DA, however it would be interesting to see whether GRACE is actually providing added value based on the variability in GRACE processing models.

Groundwater head data can be quite complex depending on the well depth and the aquifer being pumped. So far the authors only use head data without information about the aquifer systems. Different aquifer systems also result in individual specific yields. This needs to be addressed. based on a quick literature search hydrogeologic studies (e.g. Ma et al. 2005) are available for the region. Please, do provide information on whether the wells access the same aquifer. Further, groundwater heads were converted to units of storage using a scale factor (p.8, l301ff) as specific yield data were not available. Ma et al. 2005 (and probably more papers as well) provide aquifer properties for the Shiyang basin. Given that the wells are in the same aquifer system, please, show how your units of storage compare to literature values for the region.

Regarding the precipitation errors the RMS of TRMM was used (p12, l440). As the authors also compared TRMM to station data, was that error included as well?

Minor comments: - The abstract is a bit too extensive, please, shorten. - p2, l57-59. Provide reference - Fig. 1. Include all symbols in the figure caption (crosses). Since color is used, the river networks could also be added (1b). - p6, l208/209. Please, explain 'the sum of different states'. What are e.g. '4 interception' states? - p9, l331ff. What exactly was done with the NDVI values? Was the growing season length determined as the period above and below 0.2? If it was only used for visualization in Fig. 14, the section can be shortened to a couple of lines. - Fig. 14a. Is the GW head relative to amsl? What is the depth to the surface?

Sakumura, C., S. Bettadpur, and S.Bruinsma (2014), Ensemble prediction and inter-comparison analysis of GRACE time-variable gravity field models, Geophys. Res.

Lett., 41, 1389–1397,doi:10.1002/2013GL058632.

Ma J.Z., Wang X.S., and W.M. Edmunds (2005), The characteristics of groundwater resources and their changes under the impacts of human activity in the arid Northwest China — a case study of the Shiyang River Basin, Journal of Arid Environments, Volume 61, Issue 2, Pages 277-295, ISSN 0140-1963, http://dx.doi.org/10.1016/j.jaridenv.2004.07.014.

---

## Author Comment (AC1) · 9 Nov 2016

We firstly would like to acknowledge the insightful comments and suggestions provided by Dr. Bogena. Followings are the responses (R) based on the comments:

General comments concerning the hydrological modelling:

The interactive modules for simulating water abstraction etc. with the PCR-GLOBWB model are described in greater detail, but these did not be used in this study. Thus the model description should focus on the considered process.

R1: We agree with reviewer and the model description will be modified to be more concise in the revised manuscript.

[Figure]

The model parameterisation with respect to the soil hydraulic properties needs to be better described.

R2: More description of the soil properties will be added to the revised manuscript (please also see R40).

I suggest adding some plots showing the special distribution of simulated TWS for the different DA scenarios.

R3: An illustration of the spatial distribution of both DA scenarios will be shown in Fig. 11 in the revised manuscript. The following discussion will also be added in the revised manuscript: "It is also worth discussing the impact of GRACE DA on the spatial pattern of the water storage estimates. To demonstrate this, the update term ($\Delta A$ in Eq. (7)) of October 2002 from EnKF 1D and 3D cases is shown in Fig. 11. Only TWSV, SMSV, and GWSV are shown, since other components (snow, surface water, and interception) are small. As discussed above, EnKF 3D shows smaller update in all components. Due to a greater amplitude of GRACE-derived TWSV over northern and southern parts of the region (see Fig. 4), the update is mostly seen there. Almost all update is limited to the soil moisture layer. Higher precipitation is generally observed over the southern part, which leads to higher groundwater recharge (and GWSV) over that region. As such, GWSV update is clearly seen over the southern part of the region."

In the DA scheme only TWS is considered. It is no clear, how "added" or "subtracted" water was distributed by DA to the different model storages (e.g. SM, GW, snow).

R4: As only TWSV is available from GRACE, TWSV is only used in the discussion in Sect. 6.1. However, the discussion of GRACE DA impact on individual stores are given Sect. 6.2 of the revised manuscript.

Compared to the model results the variations in GRACE determined TWS are much more pronounced. Possible reasons should be discussed in greater (e.g. influence of the pattern restauration procedure).

R5: This discussion will be added to the revised manuscript as follows: "It is seen that GRACE-derived TWSV is more pronounced compared to the model estimated TWSV. This can likely be attributed to the poor quality of the model parameter calibration and the accuracy of the meteorological input data over the data-sparse regions. In the absence of observations, model parameters are difficult to determine and only the best available knowledge (or guess) is generally used, leading to inaccurate model state estimates. Updating the water storage estimates using GRACE DA showed a clear improvement in this condition."

It is unclear if at all or how groundwater abstraction was considered in the modelling. If this was considered, why was the groundwater abstraction not considered in the DA (e.g. by updating the groundwater abstraction parameter)?

R6: In this study, the state vector only contains the water storage. Groundwater abstraction is one of PCR-GLOBWB's model parameters, and it is not included in the state vector. Therefore, the groundwater abstraction is not updated or separately estimated in this study, but it is treated. Please also see R45 for more detail.

Specific comments

Title: The term "semi-arid" is not correct (see below)

R7: The "semi-arid" term is used based on Zhu et al. (2015). Please see also R10.

At times TWS variations are simply termed "TWS". This is somewhat confusing. The terms "TWS variations" or short "TWSV" should be always used.

R8: TWS variation will be changed to TWSV in the revised manuscript. Similarly, soil moisture storage variation and groundwater storage variation will be abbreviated as SMSV and GWSV.

L45: The groundwater well data should integrate of smaller areas than the catchment area of the streamflow data. Therefore, I am not convinced that this is a problem of spatial resolution.
R9: Reviewer statement is correct over a sufficiently large river basin. As GRACE spatial resolution is ∼250 km or larger, the TWSV signal of the smaller basin can be easily interfered by the neighbouring basin. This is known as a leakage effect and such an effect is seen over the Hexi Corridor. Therefore, the limited spatial resolution of GRACE plays a very important role on the state estimates there.

L57-59: According to the Köppen climate classification this region belongs to "cold desert climate" (BWk).

R10: The "semi-arid" term is used based on Zhu et al. (2015); however, we also realize that much of the region has a cold desert climate, and this can be found in the submitted manuscript (line 152): "Located next to the Gobi Desert, most parts of the region have a cold desert climate, . . ." For clarity, we will include the references of both climate classifications (Zhu et al. (2015) and Peel et al. (2007)) in section 2 of the revised manuscript.

L67-68: This depends largely on the measured variable. For instance, streamwater discharge data provides integrated information for large catchment areas.

R11: We agree with reviewer. In the revised manuscript, this sentence will be written as follows: "While streamflow gauges provide an integrated information for large catchment areas, point observations of hydrometeorological variables and even groundwater levels can be very local in scope."

L81: In addition, hydrological models typically suffer from inadequate process representations (model structure errors).

R12: The suggested statement will be added to the introduction section of the revised manuscript.

L98: "jump" of what?

R13: "jump" will be extended to "jump of the water storage estimates" in the revised manuscript.

L115: What is the size of the area?

R14: The size of the individual basin varies between 41,600 and 157,000 km2. This can be found in lines 149 - 151 of the submitted manuscript: "Shiyang River Basin (41,600 km2), the Heihe River Basin (143,000 km2), the Shule River Basin (157,000 km2), and a Desert Region (152,445 km2)"

L115-118: How do you know (e.g. the watershed area of the Rhine river is much smaller than the Hexi Corridor area)? Can you provide the SNR values for these different areas?

R15: The size of the individual basin of the Hexi Corridor is smaller than the mentioned basins, Mississippi (3,202,230 km2), Rhine (185,000 km2), Mackenzie (1,743,058 km2). The SNR values of the Hexi Corridor is approximately 2.5, compared to Mississippi (SNR $\approx$ 11), Rhine (SNR $\approx$ 17), Mackenzie (SNR $\approx$ 20).

L122: What is the difference between "surface water" and "inundated water"?

R16: The "surface water" in PCR-GLOBWB consists of river/channels, as well as lake and reservoir storages, while the term "inundated water" is conceptualized for the inundated water above the paddy field during the growing season. The terms are clearly described in PCR-GLOBWB literature (see e.g. Wada et al., 2014).

L128-129: In which way are the results validated against remote sensing?

R17: The validation is qualitatively analysed in terms of the correlation coefficient, Nash-Sutcliff coefficient, Root-Mean-Square different (RMSD). The statement will be added to the revised manuscript.

L147: The term "basin" is not appropriate.

R18: The term "basin" will be changed to "region" in the revised manuscript.

L181: "distributed hydrological model"

R19: The term "global hydrological model" will be changed to "global distributed hydrological model" in the revised manuscript.

L184-185: Also indicate the temporal resolution of the model.

R20: The statement ". . . and temporal resolution of 1 day" will be added to the revised manuscript.

L185-193: It is unclear, how or if at all these interactive modules for simulating water abstraction etc. have been used in this study. Clearly it was not the focus of this study. Thus I suggest removing this section incl. Appendix A.

R21: The section, including Appendix A, will be removed from the manuscript.

L197: Delete "an"

R22: "an" will be removed from the manuscript.

L208: Change "states" into "water storages"

R23: The term "states" will be changed to "water storage components" in the revised manuscript.

L219: This is rather a conceptual model.

R24: Reviewer is correct, like many numerical models, it is conceptual in nature.

L230: Explain "complete to the degree and order 60"

R25: The Earth gravity field is generally presented using a set of spherical harmonic coefficients (SHC) to a certain degree and order. The GRACE CSR product is provided the gravity model up to SHC degree and order 60. Therefore, we compute the TWS variation using the SHC complete to the maximum degree and order 60 in this study.

L259: Does this increment correspond to the monthly change in TWS?

R26: The increment is not necessarily (or linearly) corresponding to the filtered TWS

change. The increment rather reflects the missing signal that was caused by the filter applied. In other words, the spatial pattern of the restored TWS change (after signal restoration process applied) is not necessarily similar to the filtered TWS change (see Fig. 4a compared to Fig. 4f).

L261: Is this the general uncertainty of GRACE?

R27: Based on the previous GRACE literature (Wahr et al., 2006; Klees et al., 2008; Dahle et al., 2014), GRACE uncertainty averaged-globally is approximately 2 cm.

L263-264: By looking at Fig. 4 this procedure seems to have mainly intensified the already existing pattern. To which extent are the temporal variations in TWS estimates influenced by this procedure?

R28: The signal is generally damped after the filter is used, results in <4 cm of the TWSV amplitude (please see Fig. 4a). The signal restoration process is used to restore the mitigated signal that was caused by the filter applied. The process restores the signal back for each iteration and the TWSV amplitude becomes ∼7 cm after 6 iterations (see Fig. 4f). The spatial pattern between Fig. 4a and Fig. 4f is also different (see the contour lines). As the signal restoration process acts differently (e.g., number of iteration) for different month, the temporal variations in TWSV estimates are also influenced by this procedure. Extensive discussion of the signal restoration process can be found in the given reference (e.g., Tangdamrongsub et al., 2016).

L287-289: It is well-know that global precipitation products show considerable uncertainties, which is also indicated by the low NS values. Since in-situ data is available, I suggest to correct the TRMM data product using the approach suggested by Condom et al. (2011).

R29: We agree with reviewer that correcting TRMM using the method proposed by Condom et al. (2011) is a good idea. However, since the in situ data over the Hexi Corridor is very sparse and does not cover all model grid cells, further analysis is needed

to investigate the impact of the method on the spatial distribution. Particularly, the impact on higher frequency (daily) of the precipitation data used in this study (compared to monthly of Condom et al. (2001)). Also, there might be a chance of introducing artefacts into the TRMM data in the grid cells if no in situ data is available. Therefore, we do not apply any correction to TRMM data, and use the standard error the product provided to represent the data uncertainty.

L298: Actual or potential ET?

R30: "evapotranspiration" will be changed to "potential evapotranspiration" in the revised manuscript.

L327-329: Actually, more appropriate data is available from other gauging stations in the Hexi Corridor for this study (see e.g. Zhang et al., 2015, 2016).

R31: We thank for reviewer's information. However, we only had an access to limited ground observations by the time this study is conducted. More ground observations will be considered in future work.

L307-322: Because of this conversion method any comparison of groundwater storage changes from in-situ and GRACE observations will not be independent. This needs to be discussed in some detail. In addition, in the procedure described in Tangdamrong-sub et al. (2015) two parameter were used instead of one. Please comment on this difference.

R32: Due to the fact that the estimated scale factor values are in line with the specific yield from the field observations (please also see R33), the bias of the estimated parameter from our approach can be considered small over the Shiyang River Basin. However, we understand reviewer's concern, and therefore one additional paragraph will be added to the conclusion section as follows: "The conversion approach between the groundwater head measurement and groundwater storage is proven feasible over the Shiyang River Basin. The approach delivers comparable ranges of scale factor es-

timates to the specific yield estimated from the field observation. However, it is noted here that the results of the conducted validation might be over-optimistic, since the well data processed with the adopted conversion procedure are not fully independent of assimilated GRACE data. The specific yield from the field observation must be used when available." Additionally, the difference between 1 and 2 parameters are only the bias (first parameter, "a" parameter in Tangdamrongsub et al. (2015)) becomes very small ($\sim$1e-14) when the TWS variation and head variation are considered. Therefore, Eq. (1,2) of Tangdamrongsub et al. (2015) and Eq. (2,3) in the submitted manuscript provide the same result. However, for consistency, we restore the bias term in the revised manuscript as $\Delta GWS\_\{GRACE\text{-}\Delta SM\}+e = b+ f\cdot\Delta h$ (2) $\Delta GWS\_\{in\ situ\} = \hat{b}+ \hat{f}\cdot\Delta h$ (3)

L317-318: Please provide a figure with the data and the regression.

R33: The figure of the regression analysis is shown below (please see Fig. R1). To reduce the redundancy, we do not include Fig. R1 in the manuscript, but instead we will include a discussion of the parameter estimation in the revised manuscript as follows: "Yang et al. (2001) showed that the specific yield values obtained from the field measurements over the Shiyang River Basin was between 0.01 and 0.3. Although, the measurement was not conducted at the well stations used in this study, the values obtained can be used as a guidance of the specific yield of the Shiyang River Basin. In this study, the head measurements were converted to storage unit with the approach described in Sect. 4.3.1. The bias term in Eq. (3) was found to be very close to zero, as the variation (mean removed) was used in the regression analysis. The estimated scale factor was 0.23, 0.04, 0.24, 0.25, and 0.32 at W1 – W5, respectively, which was in line with the values obtained from the field measurement."

L320: Why are you using an averaged f value to calculate the groundwater storage for each well? I would have thought that the variations in parameter f should represent local variations in storage parameters of the aquifers. Please explain the reasoning behind this procedure.

R34: The parameter is individually estimated and used for each well location. No average parameter is used. For clarity, we will extend the statement as follows: "... and $\triangle$GWS(GRACE-$\triangle$SM) at each individual location, a bias (b), a scale factor (f) ..."

L451: Please explain how you selected these parameters (e.g. did you use a sensitivity test?).

R35: We selected these parameters based on several previous PCR-GLOBWB studies (e.g. Sutanudjaja et al., 2011, 2014), showing that these selected parameters are indeed the sensitive ones to model simulation results. For clarity, the reference will be added to the revised manuscript.

L526-527: Change into Figure 10

R36: "Fig. 9" will be changed to "Figure 10" in the revised manuscript.

L545: "on" instead of "of"

R37: "of" will be changed to "on" in the revised manuscript.

L549-550: Please provide information on the origin of these parameter values.

R38: The origin of the parameter values is given in Sutanudjaja et al. (2011, 2014), and the reference will be given in the revised manuscript. Further information related to the origin of parameter values were provided in Appendix A of the submitted manuscript. However, they are removed based on reviewer suggestion (see R21). The model parameters of PCR-GLOBWB are derived from several globally available datasets that are listed as follows. The Global Land Cover Characteristics Data Base Version 2.0 (GLCC 2.0, http://edc2.usgs.gov/glcc/globe int.php) and and FAO soil maps (1995) were used in order to parameterize the land cover and upper sub-surface properties. For mapping aquifers and estimating the groundwater recession coefficient, the GLobal HYdrogeology MaPS (GLHYMPS) global maps of permeability and porosity (Gleeson et al., 2014), as well as available global digital elevation models (e.g. HydroSHEDS, Lehner et al., 2008) were used. For further explanation about the PCR-GLOBWB

model parameterization, the reader is referred to the technical reports (e.g. van Beek and Bierkens, 2009; van Beek, 2008); and other relevant publications (e.g. Sutanudjaja et al., 2011, 2014).

L543: How do you know that the groundwater store of the Desert Region is small.

R39: We realized that the statement is misleading and we change our statement to " …the small amplitude of the groundwater variation of this region …". Small GWSV over the Desert Region is presented in Fig. 10k.

L553-554: Please explain in greater detail, why higher values of K_sat and lower values of J have led to a smaller amount of water addition.

R40: We realize that the interpretation the amount of water storage in terms of K_sat and J might be misleading as they do not have a linear relationship. Instead, soil water storage capacity (SC, see Table 1) and forcing data have greater impact on the water storage estimate. Note that greater SC value leads to greater amount of water stored in soil layer, and consequently lesser water percolate to the groundwater store. Therefore, we remove the statement related to K_sat and J, and change the analysis of this section to: "The impact of GRACE DA on different stores was influenced by both the model parameters and the forcing data assigned. The 4 basins have similar soil water storage capacities (see Table 3), which indicates that the basins can store similar amounts of soil water and generate similar amount of groundwater recharge under the same rainfall condition. However, the 4 basins received different amounts of rainfall and therefore resulted in different SMSV and GWSV estimates. For example, the Shiyang River Basin received the greatest amount of rainfall ($\sim$ twice of Heihe River Basin), which led to the greatest amount of the SMSV estimate ($\sim$1 cm annual amplitude). Such large amount was also sufficient to percolate into the groundwater layer, resulted in GWSV of $\sim$0.7 cm (see Fig. 10i and Table 2). In contrast to the Shiyang River Basin, the Desert Region received approximately 3 times less amount of rainfall, which led to a somewhat smaller amount of SMSV $\sim$0.7 cm (annual amplitude), $\sim$0.2 cm of GWSV

(see Fig. 10g, k)." The above paragraph will be added to the revised manuscript.

L599-600: I wonder whether the better agreement with the GRACE DA results is due to (or a least partly due to) the scaling procedure of the piezometer data. Please add a discussion on this.

R41: Due to the fact that the estimated scale factor values are in line with the specific yield from the field observations (please also see R33), the bias of the estimated parameter from our approach can be considered small over the Shiyang River Basin. However, we understand reviewer's concern, and therefore add one additional paragraph into the conclusion of the revised manuscript: "The conversion approach between the groundwater head measurement and groundwater storage is proven feasible over the Shiyang River Basin. The approach delivers comparable ranges of scale factor estimates to the specific yield estimated from the field observation. However, it is noted here that the results of the conducted validation might be over-optimistic, since the well data processed with the adopted conversion procedure are not fully independent of assimilated GRACE data. The specific yield from the field observation must be used when available."

L642: Clearly, predictions for G2 were improved to a lesser degree.

R42: We agree with reviewer. For clarity, the statement will be changed to "Lesser degree of improvements was observed at G2".

L647-648: These are very low amounts of precipitation, indicating very local precipitation events. It would be interesting to see the spatial distribution of these rainfall events and the resulting modelled soil moisture distribution.

R43: The maps of rainfall and SM storage estimates of the discussed events (September 2007, 2008) are shown below (please see Fig. R2). However, this is beyond the scope of this study, and therefore Fig. R2 is not presented in the manuscript.

L676-678: Why should the SM storage of the Desert Region decrease although pre-

cipitation shows an increasing trend? Please discuss.

R44: The discussion will be added to the revised manuscript: "In the Desert Region, in contrast to other basins, the minor decreasing TWS trend of -0.1 cm/yr was dominated by loss of SM storage. This was likely caused by inaccurate model parameter calibration over the Desert Region (i.e., too large SC value). Separation of the TWS into groundwater and soil moisture store was likely incorrect. As such, GRACE update was mostly attributed to the SM component, so that a groundwater pumping signature (Jiao et al., 2015) was seen in the SM instead of the GWS layer." Further discussion of this (and related) issue is also included in the conclusion: "It should be emphasized that GRACE does not fix a technical problem of the hydrological model, but it rather provides information which is not available otherwise. . . ."

L712-714: Until now, there was no indication that groundwater abstraction was considered in the modelling. Please add a description. Why was the groundwater abstraction not considered in the DA?

R45: In this study, the state vector only contains the water storage. As the groundwater abstraction is a parameter of PCR-GLOBWB, it is not included in the state vector. Therefore, the groundwater abstraction is not separately estimated in this study. However, the information of groundwater abstraction is contained in GRACE observation. Once GRACE DA is applied, such information is propagated into the state vector, particularly the groundwater layer. This is clearly seen in the negative trend of updated groundwater estimates. This discussion will be included in the conclusion of the revised manuscript. "It should be emphasized that GRACE does not fix a technical problem of the hydrological model, but it rather provides information which is not available otherwise. Note that, in principle, the model may predict any long-term behaviour of water storage, but that information should be brought in "by hand" (e.g., via the groundwater abstraction parameter). As soon as that information is not available, reliable long-term predictions on the basis of hydrological modelling alone are conceptually impossible."

L734-735: See comment above. Would it be possible to update the groundwater abstraction parameter?

R46: Yes, it is possible to update the model parameter together with the state vector. We will consider reviewer's suggestion in the future work.

L744: Please provide quantitative information on groundwater abstraction.

R47: As the groundwater abstraction is not estimated by our GRACE DA approach, we do not quantify the amount of groundwater abstraction in this study. The groundwater abstraction can be quantified when the parameter is estimated together with the state vector.

References

Dahle, C., Flechtner, F., Gruber, C., König, D., König, R., Michalak, G., and Neumayer, K.-H.: GFZ RL05: An Improved Time-Series of Monthly GRACE Gravity Field Solutions, In Flechtner, F., Sneeuw, N., Schuh, W.-D. (Eds.), Observation of the System Earth from Space - CHAMP, GRACE, GOCE and future missions, (GEOTECHNOLOGIEN Science Report; 20; Advanced Technologies in Earth Sciences), Berlin, Springer, 29-39, http://doi.org/10.1007/978-3-642-32135-1_4, 2014.

Klees, R., Liu, X., Wittwe, T., Gunter, B. C., Revtova, E. A., Tenzer, R., Ditmar, P., Winsemius, H. C., and Savenije, H. H. G.: A Comparison of Global and Regional GRACE Models for Land Hydrology, Surv. Geophys., 29, 335-359, doi:10.1007/s10712-008-9049-8, 2008.

Peel, M. C., Finlayson, B. L., and McMahon, T. A.: Updated would map of the Köppen-Geiger climate classification, Hydrol. Earth Syst. Sci., 11, 1633–1644, 2007.

Sutanudjaja, E. H., van Beek, L. P. H., de Jong, S. M., van Geer, F. C., and Bierkens, M. F. P.: Large-scale groundwater modeling using global datasets: a test case for the {Rhine-Meuse} basin, Hydrol. Earth Syst. Sci., 15(9), 2913–2935, doi:10.5194/hess-15-2913-2011, 2011.

Sutanudjaja, E. H., van Beek, L. P. H., de Jong, S. M., van Geer, F. C., and Bierkens, M. F. P.: Calibrating a large-extent high-resolution coupled groundwater-land surface model using soil moisture and discharge data. Water Resour. Res., 50, 687–705. doi:10.1002/2013WR013807, 2014.

Tangdamrongsub, N., Ditmar, P. G., Steele-Dunne, S. C., Gunter, B. C., and Sutanudjaja, E. H.: Exploring irregular flood events over Tonlé Sap basin in Cambodia using GRACE and MODIS satellite observations combined with altimetry observation and hydrological models, Remote Sens. Environ., 181, 162 – 173, http://dx.doi.org/10.1016/j.rse.2016.03.030, 2016.

Wada, Y., Wisser, D., and Bierkens, M. F. P.: Global modeling of withdrawal, allocation and consumptive use of surface water and groundwater resources. Earth System Dynamics, 5, 15–40. doi:10.5194/esd-5-15-2014, 2014.

Wahr, J., Swenson, S., and Velicogna, I.: Accuracy of GRACE mass estimates, Geophys. Res. Lett., 33, L06401, doi:10.1029/2005GL025305, 2006.

Zhu, J. F., Winter, C. L., and Wang Z. J.: Nonlinear effects of locally heterogeneous hydraulic conductivity fields on regional stream–aquifer exchanges, Hydrol. Earth Syst. Sci., 19, 4531–4545, 2015.

[Figure]

[Figure]

**Fig. 1.** Figure R1. Regression analysis between GRACE-GLDAS and adjusted well measurements in 5 different locations.

[Figure]

**Fig. 2.** Figure R2. Monthly total precipitation (left) and SM storage estimates of September 2007 and 2008. Stream gauge location G2 is also shown.

---

## Author Comment (AC2) · 9 Nov 2016

We firstly would like to acknowledge the insightful comments and suggestions provided by reviewer 2. Followings are the responses (R) based on the comments:

Different GRACE gravity field models are available, CSR (this study, p6, l227ff), GFZ, JPL, CNES/GRGS (Sakumura et al 2014). Why was CSR selected and how are the differences between the different GRACE processing models for the study region. I understand that the focus of the article is on the added valued of the DA, however it would be interesting to see whether GRACE is actually providing added value based on the variability in GRACE processing models.

R1: Comparing to GFZ, JPL, and CNES/GRGS, the CSR product is the only product that provides the error variance covariance matrix of the spherical harmonic coefficients. Therefore, it is selected in this study. Note here that the variance covariance matrix is the only information that reflects the true GRACE error. As this information is not available from GFZ, JPL, and CNES/GRGS, they are not considered in this study. We agree with reviewer that it would be interesting to see whether GRACE is consistently improving the water storage estimates based on different products used. The comparison can be conducted as soon as the error information from other data centre is released.

Groundwater head data can be quite complex depending on the well depth and the aquifer being pumped. So far the authors only use head data without information about the aquifer systems. Different aquifer systems also result in individual specific yields. This needs to be addressed. based on a quick literature search hydrogeologic studies (e.g. Ma et al. 2005) are available for the region. Please, do provide information on whether the wells access the same aquifer. Further, groundwater heads were converted to units of storage using a scale factor (p.8, l301ff) as specific yield data were not available. Ma et al. 2005 (and probably more papers as well) provide aquifer properties for the Shiyang basin. Given that the wells are in the same aquifer system, please, show how your units of storage compare to literature values for the region.

R2: We thank reviewer 2 for this valuable information. Unfortunately, the data we used does not come with the aquifer information, so we cannot guarantee whether the well accesses the same aquifer as in Ma et al. (2005). As such, the specific yield is computed based on the best hydrological knowledge (model) and observation. The estimated values are between 0.04 and 0.3, which is in line with the specific yield values Yang et al. (2001) determined from the pumping tests, $0.01 - 0.3$. Therefore, our estimate value can be considered sufficiently accurate for the head conversion. For clarity, we add the additional statement to the revised manuscript: "Yang et al. (2001) showed that the specific yield values obtained from the field measurements over the

Shiyang River Basin was between 0.01 and 0.3. Although, the measurement was not conducted at the well stations used in this study, the values obtained can be used as a guidance of the specific yield of the Shiyang River Basin. In this study, the head measurements were converted to storage unit with the approach described in Sect. 4.3.1. The bias term in Eq. (3) was found to be very close to zero, as the variation (mean removed) was used in the regression analysis. The estimated scale factor was 0.23, 0.04, 0.24, 0.25, and 0.32 at W1 – W5, respectively, which was in line with the values obtained from the field measurement."

Regarding the precipitation errors the RMS of TRMM was used (p12, l440). As the authors also compared TRMM to station data, was that error included as well?

R3: As the error of other precipitation products are not available, no error is included in the analysis of Sect. 4.2 to avoid the inconsistency of the comparison.

Minor comments: The abstract is a bit too extensive, please, shorten.

R4: The abstract will be shortened in the revised manuscript.

p2, l57-59. Provide reference

R5: References (Gong et al., 2004; Zhu et al., 2015; Cui and Shao, 2005) will be given in the revised manuscript.

Fig. 1. Include all symbols in the figure caption (crosses). Since color is used, the river networks could also be added (1b).

R6: The symbol will be added to Fig. 1 caption of the revised manuscript as " . . .the locations of considered groundwater wells (x) and river stream gauges (+)." The river network will also be added to Fig. 1b.

p6, l208/209. Please, explain 'the sum of different states'. What are e.g. '4 interception' states?

R7: TWS variation is computed from the sum of 27 different water storage components

(layers), which are 8 soil moisture layers, 2 groundwater layers, 4 interception layers, 8 snow layers, 4 inundated top water layers, and 1 surface water layer. For clarity, we revise the statement to:

". . . the total water storage (TWS) is computed as the sum of 27 different water storage components: 8 soil moisture layers, 2 groundwater layers, 4 interception layers, 8 snow layers, 4 inundated top water layers, and 1 surface water layer.".

p9, l331ff. What exactly was done with the NDVI values? Was the growing season length determined as the period above and below 0.2? If it was only used for visualization in Fig. 14, the section can be shortened to a couple of lines.

R8: NDVI and GWS variation were analysed together to determine if the growing season was being extended beyond the limited rainy period through groundwater extraction for irrigation. The reviewer is correct in that the growing season length is determined as the period above ∼0.2. In the revised manuscript, we remove a few statements in Sect. 4.4.3 to make the section more concise.

Fig. 14a. Is the GW head relative to amsl? What is the depth to the surface?

R9: Yes, the measurement is relative to the mean sea level. For clarity, we will add an additional statement to the revise manuscript:

". . . form of piezometric heads (relative to the mean sea level), . . ."

The depth from to the surface is not available from the data provider, and therefore we cannot provide the value here.

References

Cui, Y. and Shao, J.: The Role of Ground Water in Arid/Semiarid Ecosystems, Northwest China, Groundwater, 43 (4), 471–477, doi:10.1111/j.1745-6584.2005.0063.x, 2005.

Gong, D. Y., Shi, P. J., and Wang, J. A.: Daily precipitation changes in the

semi-arid region over northern China. J. Arid. Environ., 59 (4), 771–784, doi:10.1016/j.jaridenv.2004.02.006, 2004.

Zhu, J. F., Winter, C. L., and Wang Z. J.: Nonlinear effects of locally heterogeneous hydraulic conductivity fields on regional stream–aquifer exchanges, Hydrol. Earth Syst. Sci., 19, 4531–4545, 2015.

---

## Short Comment (SC1) · 5 Dec 2016

I read the manuscript by Tangdamrongsub et al. with an interest. The manuscript addresses important methodological issues of GRACE data assimilation into hydrological models. There are, however, serious issues in the methodological implementation of this paper (especially the treatment of the observation vector in Eq. (7)). I also found a number of examples in which previous studies are not correctly referred (in terms of the details of methodology). My very relevant previous work on modeling errors of GRACE TWS changes is ignored. I believe that the following comments are important to be addressed by the authors prior to the publication of the manuscript in a prestigious journal like HESS.

Major comments on methodology:

1) Treating observations as random variable

l. 368 and matrix D in Eq. (7): Burgers et al. (1998) showed that it is necessary to consider the observations as random variable, i.e. that not only an ensemble of predicted model states but also an ensemble of observations has to be considered when calculating the update of each model ensemble member. Perturbations for the observations can be drawn from the error covariance matrix R. Otherwise, the error statistics of the updated model ensemble are underestimated (i.e. not correctly treated). In a correct implementation, matrix D does not contain N identical columns as described in l. 368. This should be fixed or at least discussed by the authors. In addition, it is not possible to draw random errors from the full error covariance matrix of GRACE TWS changes on a 0.5x0.5 degree grid, since the matrix has a rank deficiency. This is a critical issue and should be addressed by the authors as well.

l. 507-508: The standard deviations of the EnKF results are however underestimated, since the observation vector was not treated as a random variable in Eq. (7). Therefore, the error statistics of the updated model states are not correct. This should be fixed or at least discussed.

l. 588-589: This might change after correctly estimating the updated model ensemble spread by generating perturbations for the observations (revising Eq. (7)).

2) Characteristics of error covariance matrices

Eq. (8): Since both error covariance matrices (from the model and the observations) have a rank-defect due to (1) the fact that usually the number of model states is much larger than the number of model ensemble members and (2) GRACE cannot actually resolve TWS changes on a 0.5x0.5 degree grid, the inverse in Eq. (8) does not exist. This should be pointed out and a reference to sections 5.2.1 and 5.2.2 might be provided that describe how the authors deal with this issue.

[Figure]

l. 251: GRACE observations are highly correlated on such a fine spatial resolution (similar to the above comment). Did the authors investigate this? Was this the reason to use a maximum correlation length for the observation error covariance matrix?

l. 414-415: If I understand it correctly, the error correlation length is set to 250 km and TWS changes outside of this radius are assumed to not be correlated to the center grid cell. Is this reasonable? It would be helpful to investigate the correlations of points with longer distances to verify this choice. Does the "local" error covariance matrix have a full rank?

Fig. 7: In the main text (l. 414-415), it is explained that a correlation length of 250 km is used (approx. four to five 0.5x0.5 degree ($\sim$50kmx50km at the equator) grid cells in each direction from the center grid cell). In Fig. 7, it is shown that only the neighboring grid cells are considered. Please clarify.

l. 419: Since the neighboring 0.5x0.5 degree grid cells are highly correlated, it is not reasonable - based on the GRACE error characteristics - to apply the EnKF without spatial error correlations on such a fine scale. A statement would be helpful to the reader.

l. 726-727: But: The authors do not use the full error covariance matrix as directly calculated from the observations. Instead a maximum correlation length of 250 km is assumed, and thus a part of the information within the full error covariance matrix is neglected. Therefore, the statement might be misleading.

Major comments on citation of previous works:

3) Zaitchik et al. (2008)

l. 90-91: That seems to be incorrect. Zaitchik et al. (2008) used an ensemble Kalman smoother (EnKS) approach to partition the monthly update increment (based on comparing monthly means of modeled and observed TWS changes) equally to each day of the month. GRACE TWS changes are only assimilated once per month and not every

10 days.

4) Forman et al. (2012)

l. 95: This work adapts the method as proposed in Zaitchik et al. (2008) to a snow-dominated basin.

l. 98: Please also consider the disadvantage of computational costs: The method has some computational drawback since the model has to be evaluated twice over the same month.

5) Forman et al. (2013)

l. 106: In Forman et al. (2013), the authors did not use correlated errors for the data assimilation. They investigated for which spatial resolution errors of GRACE TWS changes might be considered as uncorrelated. According to these investigations, they assumed white noise for (sub-)basin averaged TWS changes from GRACE.

6) Girotto et al. (2016)

l. 89-95: In this work, the authors performed an analysis of introducing the update increments completely at the beginning of a month, the end of a month or equally distributed over all days of a month. This is worth to be mentioned along with the other citations.

7) Schumacher et al. (2016)

l. 39-40 and l. 106-108: A first analysis of assessing the effect of considering or neglecting spatial error correlations of GRACE TWS changes was performed in Schumacher et al. (2016) in form of a synthetic experiment, for which one of the authors of this HESSD manuscript was the editor and should therefore be very familiar with the work. It seems that the paper is methodologically the closest related to the analysis presented here and, therefore, should be cited and discussed. Findings should be compared to the findings in the published paper.

l. 577: This was also seen and discussed in Schumacher et al. (2016). The authors should compare their results with the findings in this paper, since the objective of both papers is to understand the effect of considering spatial error correlations of GRACE TWS changes on hydrological data assimilation results.

l. 715-718: The authors should add something like "in agreement with the recommendation in Schumacher et al. (2016)."

l. 719-724: The findings in the HESSD manuscript allow for a clearer conclusion on improvements when error correlations of GRACE TWS changes are taken into account. What might be the reason for this? Differences in the study set up? Localization of model / observation error covariance matrices?

l. 729: A reference to Schumacher et al. (2016) would strengthen this statement, since the HESSD manuscript is not the only study that concludes a benefit / more realistic GRACE data assimilation approach if implementing GRACE error correlations.

l. 752-753: Schumacher et al. (2016) should be added to the list of references.

l. 755: Alternative methods have been investigated in Schumacher et al. (2016), namely a square root analysis scheme (SQRA) and the singular evolutive interpolated Kalman filter (SEIK). Especially the application of the SEIK filter showed promising results. A citation would support the authors expectation that alternative methods, e.g. the particle filter, would improve the data assimilation performance.

Minor comments:

l. 583: "truth", i.e. to the independent measurements of individual water compartments. These measurements are also subject to uncertainties and not "true" values.

l. 756: "true" -> better "full" (true is difficult since often unknown / poorly known)

References:

Burgers, G., P.J. van Leeuwen and G. Evensen (1998). Analysis scheme in the

ensemble Kalman filter. Mon Weather Rev, 126:1719–1724. doi:10.1175/1520-0493(1998)126<1719:ASITEK>2.0.CO;2.

Girotto, M., G.J.M. De Lannoy, R.H. Reichle and M. Rodell (2016). Assimilation of gridded terrestrial water storage observations from GRACE into a land surface model. Water Resour Res, 52(5):4164–4183. doi:10.1002/2015WR018417.

Schumacher, M., J. Kusche and P. Döll (2016). A Systematic Impact Assessment of GRACE Error Correlation on Data Assimilation in Hydrological Models. J Geod, 90(6): 537–559. doi:10.1007/s00190-016-0892-y.

---

## Author Comment (AC3) · 26 Jan 2017

We firstly would like to acknowledge the insightful comments and suggestions provided by M. Schumacher. Followings are the responses (R) based on the comments:

1) Treating observations as random variable l. 368 and matrix D in Eq. (7): Burgers et al. (1998) showed that it is necessary to consider the observations as random variable, i.e. that not only an ensemble of predicted model states but also an ensemble of observations has to be considered when calculating the update of each model ensemble member. Perturbations for the observations can be drawn from the error covariance matrix R. Otherwise, the error statistics of the updated model ensemble are underestimated (i.e. not correctly treated). In a correct implementation, matrix D does not

contain N identical columns as described in l. 368. This should be fixed or at least discussed by the authors.

R1: We implemented the EnKF as outlined by Evensen (2003). In our formulation, D contains the perturbed observations, i.e. each column is a replicate of the observation but perturbed with $\sim$N(0,R). This was not articulated well in the previous version of the manuscript. The text will be corrected to make this clearer as follows: "the GRACE observation vector is stored in the matrix D_{mxN}, in which each column is a replicate of the observation but perturbed with random noise $\sim$N(0,R). The analysis equation can be expressed as (Evensen, 2003): "

In addition, it is not possible to draw random errors from the full error covariance matrix of GRACE TWS changes on a 0.5x0.5 degree grid, since the matrix has a rank deficiency. This is a critical issue and should be addressed by the authors as well.

R2: In our study, the error variance-covariance matrix associated with the post-processed GRACE data was used. We did not use the original error matrix since it did not represent the filtered GRACE signal used in our study. In our covariance computation (described in Sect. 5.2.2), the localization function with correlation length similar to the Gaussian smoothing used was applied. Although the main objective of the covariance localization is to reduce the spurious correlation at long distance caused by the limited realization number, the localization also affects the correlation at short distance, and a strong correlation at a short distance becomes slightly weaker. As a result, the error variance-covariance matrix derived based on our method has a full rank. Applying localization also improved the condition number of the covariance matrix, e.g., from $\sim$10^{14} to $\sim$10^{2} found in our study. Similar to Eicker et al. (2014), the matrix rank and condition number were determined using Matlab functions rank and cond, respectively. We thank reviewer for the advice. The clarification regarding rank deficiency will be included in the revised manuscript.

l. 507-508: The standard deviations of the EnKF results are however underestimated,
since the observation vector was not treated as a random variable in Eq. (7). Therefore, the error statistics of the updated model states are not correct. This should be fixed or at least discussed.

R3: Please see R1

l. 588-589: This might change after correctly estimating the updated model ensemble spread by generating perturbations for the observations (revising Eq. (7)).

R4: Please see R1

2) Characteristics of error covariance matrices Eq. (8): Since both error covariance matrices (from the model and the observations) have a rank-defect due to (1) the fact that usually the number of model states is much larger than the number of model ensemble members and (2) GRACE cannot actually resolve TWS changes on a 0.5x0.5 degree grid, the inverse in Eq. (8) does not exist. This should be pointed out and a reference to sections 5.2.1 and 5.2.2 might be provided that describe how the authors deal with this issue.

R5: Please see R2

l. 251: GRACE observations are highly correlated on such a fine spatial resolution (similar to the above comment). Did the authors investigate this? Was this the reason to use a maximum correlation length for the observation error covariance matrix?

R6: Reviewer is correct. In our covariance computation (described in Sect. 5.2.2), the localization function with correlation length similar to the Gaussian smoothing used was applied. The localization helps to improve the matrix stability and we investigated this by checking the rank and condition number of the matrix as explained in R2.

l. 414-415: If I understand it correctly, the error correlation length is set to 250 km and TWS changes outside of this radius are assumed to not be correlated to the center grid cell. Is this reasonable? It would be helpful to investigate the correlations of points with longer distances to verify this choice. Does the "local" error covariance matrix have a

full rank?

R7: As the observation error variance-covariance matrix is derived based on the application of 250 km filter radius, the correlation error at distance beyond 250 km (correlation length) does not have a crucial impact on the result. In the submitted manuscript, we demonstrate the error characteristic in Fig. 8b. From the figure, the correlation reduces significantly beyond the correlation length. Additionally, the error variance-covariance matrix derived based on our method has a full rank.

Fig. 7: In the main text (l. 414-415), it is explained that a correlation length of 250 km is used (approx. four to five 0.5x0.5 degree (âĹij50kmx50km at the equator) grid cells in each direction from the center grid cell). In Fig. 7, it is shown that only the neighboring grid cells are considered. Please clarify.

R8: Reviewer is correct. We realized that the figure caption was not explained clearly. To clarify this, we add an additional description in the figure caption as follows: "The graphic demonstrates the case of 1 pixel (0.5 degree) correlation distance. The boundary stretches farther for larger correlation distance."

l. 419: Since the neighboring 0.5x0.5 degree grid cells are highly correlated, it is not reasonable - based on the GRACE error characteristics - to apply the EnKF without spatial error correlations on such a fine scale. A statement would be helpful to the reader.

R9: We thank reviewer for the recommendation, the statement will be added to the conclusion section of the revised manuscript as follows: "This is likely due to the fact that the neighboring 0.5ox0.5o grid cells are highly correlated, and it is reasonable to apply the EnKF with spatial error correlations on such a fine scale."

l. 726-727: But: The authors do not use the full error covariance matrix as directly calculated from the observations. Instead a maximum correlation length of 250 km is assumed, and thus a part of the information within the full error covariance matrix is

neglected. Therefore, the statement might be misleading.

R10: We thank for reviewer comment. To clarify this, we will modify the statement in the revised manuscript as follows: "...this is a reasonable price to pay as deriving the error variance-covariance matrix from the full (and only full) error covariance matrix reflects a better representation of the real GRACE uncertainty."

Major comments on citation of previous works: 3) Zaitchik et al. (2008) l. 90-91: That seems to be incorrect. Zaitchik et al. (2008) used an ensemble Kalman smoother (EnKS) approach to partition the monthly update increment (based on comparing monthly means of modeled and observed TWS changes) equally to each day of the month. GRACE TWS changes are only assimilated once per month and not every 10 days.

R11: We thank for reviewer comment. The statement will be corrected in the revised manuscript as follows: "...using a monthly observation value and distributing the up-date as daily increments (Zaitchik et al., 2008; Forman et al., 2012)."

4) Forman et al. (2012) l. 95: This work adapts the method as proposed in Zaitchik et al. (2008) to a snow-dominated basin.

R12: Please see R11.

l. 98: Please also consider the disadvantage of computational costs: The method has some computational drawback since the model has to be evaluated twice over the same month.

R13: We thank for reviewer suggestion. The additional sentence will be added in the revised manuscript as follows: "Another disadvantage is the additional computational cost of running the model twice for the same month."

5) Forman et al. (2013) l. 106: In Forman et al. (2013), the authors did not use correlated errors for the data assimilation. They investigated for which spatial resolution errors of GRACE TWS changes might be considered as uncorrelated. According to

these investigations, they assumed white noise for (sub-)basin averaged TWS changes from GRACE.

R14: We agree with reviewer. Forman et al. (2013) will be removed in this context to avoid the confusion.

6) Girotto et al. (2016) l. 89-95: In this work, the authors performed an analysis of introducing the update increments completely at the beginning of a month, the end of a month or equally distributed over all days of a month. This is worth to be mentioned along with the other citations.

R15: We thank for reviewer suggestion. Girotto et al. (2016) will be cited in the revised manuscript.

7) Schumacher et al. (2016) l. 39-40 and l. 106-108: A first analysis of assessing the effect of considering or neglecting spatial error correlations of GRACE TWS changes was performed in Schumacher et al. (2016) in form of a synthetic experiment, for which one of the authors of this HESSD manuscript was the editor and should therefore be very familiar with the work. It seems that the paper is methodologically the closest related to the analysis presented here and, therefore, should be cited and discussed. Findings should be compared to the findings in the published paper.

R16: At the time this study was conducted, Schumacher et al. (2016) was not published, therefore we conducted the analysis independently based on our method (proposed in this HESSD paper). However, we thank reviewer for the recommendation, and Schumacher et al. (2016) will be cited in the revised manuscript.

l. 577: This was also seen and discussed in Schumacher et al. (2016). The authors should compare their results with the findings in this paper, since the objective of both papers is to understand the effect of considering spatial error correlations of GRACE TWS changes on hydrological data assimilation results.

R17: We thank for reviewer suggestion. The additional statement will be included in

the revised manuscript as follows: "The finding is somewhat in line with results from the numerical study by Schumacher et al. (2016) that considering correlated observation errors does not necessarily lead to a better agreement with GRACE observation."

l. 715-718: The authors should add something like "in agreement with the recommendation in Schumacher et al. (2016)."

R18: We thank for reviewer suggestion. The given statement will be considered in the revised manuscript.

l. 719-724: The findings in the HESSD manuscript allow for a clearer conclusion on improvements when error correlations of GRACE TWS changes are taken into account. What might be the reason for this? Differences in the study set up? Localization of model / observation error covariance matrices?

R19: The improvement is mainly due to a better representation of GRACE information in the EnKF. Ignoring error correlations in the DA led to an over-fit of the results to the observations, which led to less accurate state estimates. These statements will be presented in the revised manuscript.

l. 729: A reference to Schumacher et al. (2016) would strengthen this statement, since the HESSD manuscript is not the only study that concludes a benefit / more realistic GRACE data assimilation approach if implementing GRACE error correlations.

R20: We thank for reviewer for the suggestion, Schumacher et al. (2016) will be cited in the relevant context.

l. 752-753: Schumacher et al. (2016) should be added to the list of references.

R21: Schumacher et al. (2016) will be added to the list of references.

l. 755: Alternative methods have been investigated in Schumacher et al. (2016), namely a square root analysis scheme (SQRA) and the singular evolutive interpolated Kalman filter (SEIK). Especially the application of the SEIK filter showed promising

results. A citation would support the authors expectation that alternative methods, e.g. the particle filter, would improve the data assimilation performance.

R22: We thank the review for the suggestion. We will consider this in the revision.

Minor comments: l. 583: "truth", i.e. to the independent measurements of individual water compartments. These measurements are also subject to uncertainties and not "true" values.

R23: To avoid the confusion, the statement will be changed to: "Validating against the in situ groundwater and streamflow data will quantitatively reveal the performance of each approach"

l. 756: "true" -> better "full" (true is difficult since often unknown / poorly known)

R24: "true" will be changed to "realistic".

---

## Author Response (AR1)

**Response to reviewer 1**

*The interactive modules for simulating water abstraction etc. with the PCR-GLOBWB model are described in greater detail, but these did not be used in this study. Thus the model description should focus on the considered process.*

**R1:** We agree with reviewer and the model description will be modified to be more concise in the revised manuscript.

*The model parameterisation with respect to the soil hydraulic properties needs to be better described.*

**R2:** More description of the soil properties will be added to the revised manuscript (please also see R40).

*I suggest adding some plots showing the special distribution of simulated TWS for the different DA scenarios.*

**R3:** An illustration of the spatial distribution of both DA scenarios will be shown in Fig. 11. The following discussion will also be added in the revised manuscript **lines 644 – 653**:

"It is also worth discussing the impact of GRACE DA on the spatial pattern of the water storage estimates. To demonstrate this, the update term ($\Delta A$ in Eq. (7)) of October 2002 from EnKF 1D and 3D cases is shown in Fig. 11. Only TWSV, SMSV, and GWSV are presented, since other components (snow, surface water, and interception) are small. As discussed above, EnKF 3D shows smaller update in all components. Due to a greater amplitude of GRACE-derived TWSV over northern and southern parts of the region (see Fig. 4), the update is mostly seen there. Almost all update is limited to the soil moisture layer. Higher precipitation is generally observed over the southern part, which leads to higher groundwater recharge (and GWSV) over that region. As such, a GWSV update is clearly seen over the southern part of the region."

*In the DA scheme only TWS is considered. It is no clear, how "added" or "subtracted" water was distributed by DA to the different model storages (e.g. SM, GW, snow).*

**R4:** As only TWSV is available from GRACE, TWSV is only used in the discussion in Sect. 6.1. However, the discussion of GRACE DA impact on individual stores are given Sect. 6.2 of the revised manuscript.

*Compared to the model results the variations in GRACE determined TWS are much more pronounced. Possible reasons should be discussed in greater (e.g. influence of the pattern restauration procedure).*

**R5:** This discussion will be added to the revised manuscript **lines 509 – 515**:

"It is seen that GRACE-derived TWSV has a greater annual amplitude compared to the model estimated TWSV. This can likely be attributed to the poor quality of the model parameter calibration and the accuracy of the meteorological input data over the data-sparse regions. In the absence of observations, model parameters are difficult to determine and only the best available knowledge (or guess) is generally used, leading to inaccurate model state estimates. Updating the water storage estimates using GRACE DA showed a clear improvement in this case."

*It is unclear if at all or how groundwater abstraction was considered in the modelling. If this was considered, why was the groundwater abstraction not considered in the DA (e.g. by updating the groundwater abstraction parameter)?*

**R6:** In this study, the state vector only contains the water storage. Groundwater abstraction is one of PCR-GLOBWB's model parameters, and it is not included in the state vector. Therefore, the groundwater abstraction is not updated or separately estimated in this study, but it is treated. Please also see R45 for more detail.

*Title: The term "semi-arid" is not correct (see below)*

**R7:** The "semi-arid" term is used based on Zhu et al. (2015). Please see also R10.

*At times TWS variations are simply termed "TWS". This is somewhat confusing. The terms "TWS variations" or short "TWSV" should be always used.*

**R8:** TWS variation will be changed to TWSV in the revised manuscript. Similarly, soil moisture storage variation and groundwater storage variation will be abbreviated as SMSV and GWSV.

*L45: The groundwater well data should integrate of smaller areas than the catchment area of the streamflow data. Therefore, I am not convinced that this is a problem of spatial resolution.*

**R9:** Reviewer statement is correct over a sufficiently large river basin. As GRACE spatial resolution is ~250 km or larger, the TWSV signal of the smaller basin can be easily interfered by the neighbouring basin. This is known as a leakage effect and such an effect is seen over the Hexi Corridor. Therefore, the limited spatial resolution of GRACE plays a very important role on the state estimates there.

*L57-59: According to the Köppen climate classification this region belongs to "cold desert climate" (BWk).*

**R10:** The "semi-arid" term is used based on Zhu et al. (2015); however, we also realize that much of the region has a cold desert climate, and this can be found in the submitted manuscript (line 152):

"Located next to the Gobi Desert, most parts of the region have a cold desert climate, …"

For clarity, we will include the references of both climate classifications (Zhu et al. (2015) and Peel et al. (2007)) in section 2 of the revised manuscript.

*L67-68: This depends largely on the measured variable. For instance, streamwater discharge data provides integrated information for large catchment areas.*

**R11:** We agree with reviewer. In the revised manuscript **lines 68 – 70**, this sentence will be written as follows:

"While streamflow gauges provide integrated information for large catchment areas, point observations of hydrometeorological variables and even groundwater levels can be very local in scope."

*L81: In addition, hydrological models typically suffer from inadequate process representations (model structure errors).*

**R12:** The suggested statement will be added to the introduction section of the revised manuscript **lines 85 – 86**.

*L98: "jump" of what?*

**R13:** "jump" will be extended to "jump of the water storage estimates" in the revised manuscript **line 101**.

*L115: What is the size of the area?*

**R14:** The size of the individual basin varies between 41,600 and 157,000 km². This can be found in lines 149 - 151 of the submitted manuscript:

"Shiyang River Basin (41,600 km2), the Heihe River Basin (143,000 km2), the Shule River Basin (157,000 km2), and a Desert Region (152,445 km2)"

*L115-118: How do you know (e.g. the watershed area of the Rhine river is much smaller than the Hexi Corridor area)? Can you provide the SNR values for these different areas?*

**R15:** The size of the individual basin of the Hexi Corridor is smaller than the mentioned basins, Mississippi (3,202,230 km2), Rhine (185,000 km2), Mackenzie (1,743,058 km2). The SNR values of the Hexi Corridor is approximately 2.5, compared to Mississippi (SNR ≈ 11), Rhine (SNR ≈ 17), Mackenzie (SNR ≈ 20).

*L122: What is the difference between "surface water" and "inundated water"?*

**R16:** The "surface water" in PCR-GLOBWB consists of river/channels, as well as lake and reservoir storages, while the term "inundated water" is conceptualized for the inundated water above the paddy field during the growing season. The terms are clearly described in PCR-GLOBWB literature (see e.g. Wada et al., 2014).

*L128-129: In which way are the results validated against remote sensing?*

**R17:** The validation is qualitatively analysed in terms of the correlation coefficient, Nash-Sutcliff coefficient, Root-Mean-Square different (RMSD). The statement will be added to the revised manuscript **lines 131 – 133**.

*L147: The term "basin" is not appropriate.*

**R18:** The term "basin" will be changed to "region" in the revised manuscript **line 145**.

*L181: "distributed hydrological model"*

**R19:** The term "global hydrological model" will be changed to "global distributed hydrological model" in the revised manuscript **line 179**.

*L184-185: Also indicate the temporal resolution of the model.*

**R20:** The statement "… and temporal resolution of 1 day" will be added to the revised manuscript **lines 183 – 184**.

*L185-193: It is unclear, how or if at all these interactive modules for simulating water abstraction etc. have been used in this study. Clearly it was not the focus of this study. Thus I suggest removing this section incl. Appendix A.*

**R21:** The section, including Appendix A, will be removed from the manuscript.

*L197: Delete "an"*

**R22:** "an" will be removed from the manuscript.

*L208: Change "states" into "water storages"*

**R23:** The term "states" will be changed to "water storage components" in the revised manuscript **line 198**.

*L219: This is rather a conceptual model.*

**R24:** Reviewer is correct, like many numerical models, it is conceptual in nature.

*L230: Explain "complete to the degree and order 60"*

**R25:** The Earth gravity field is generally presented using a set of spherical harmonic coefficients (SHC) to a certain degree and order. The GRACE CSR product is provided the gravity model up to SHC degree and order 60. Therefore, we compute the TWS variation using the SHC complete to the maximum degree and order 60 in this study.

*L259: Does this increment correspond to the monthly change in TWS?*

**R26:** The increment is not necessarily (or linearly) corresponding to the filtered TWS change. The increment rather reflects the missing signal that was caused by the filter applied. In other words, the spatial pattern of the restored TWS change (after signal restoration process applied) is not necessarily similar to the filtered TWS change (see Fig. 4a compared to Fig. 4f).

*L261: Is this the general uncertainty of GRACE?*

**R27:** Based on the previous GRACE literature (Wahr et al., 2006; Klees et al., 2008; Dahle et al., 2014), GRACE uncertainty averaged-globally is approximately 2 cm.

*L263-264: By looking at Fig. 4 this procedure seems to have mainly intensified the already existing pattern. To which extent are the temporal variations in TWS estimates influenced by this procedure?*

**R28:** The signal is generally damped after the filter is used, results in <4 cm of the TWSV amplitude (please see Fig. 4a). The signal restoration process is used to restore the mitigated signal that was caused by the filter applied. The process restores the signal back for each iteration and the TWSV amplitude becomes ~7 cm after 6 iterations (see Fig. 4f). The spatial pattern between Fig. 4a and Fig. 4f is also different (see the contour lines). As the signal restoration process acts differently (e.g., number of iteration) for different month, the temporal variations in TWSV estimates are also influenced by this procedure. Extensive discussion of the signal restoration process can be found in the given reference (e.g., Tangdamrongsub et al., 2016).

*L287-289: It is well-know that global precipitation products show considerable uncertainties, which is also indicated by the low NS values. Since in-situ data is available, I suggest to correct the TRMM data product using the approach suggested by Condom et al. (2011).*

**R29:** We agree with reviewer that correcting TRMM using the method proposed by Condom et al. (2011) is a good idea. However, since the in situ data over the Hexi Corridor is very sparse and does not cover all model grid cells, further analysis is needed to investigate the impact of the method on the spatial distribution. Particularly, the impact on higher frequency (daily) of the precipitation data used in this study (compared to monthly of Condom et al. (2001)). Also, there might be a chance of introducing artefacts into the TRMM data in the grid cells if no in situ data is available. Therefore, we do not apply any correction to TRMM data, and use the standard error the product provided to represent the data uncertainty.

*L298: Actual or potential ET?*

**R30:** "evapotranspiration" will be changed to "potential evapotranspiration" in the revised manuscript **line 287**.

*L327-329: Actually, more appropriate data is available from other gauging stations in the Hexi Corridor for this study (see e.g. Zhang et al., 2015, 2016).*

**R31:** We thank for reviewer's information. However, we only had an access to limited ground observations by the time this study is conducted. More ground observations will be considered in future work.

*L307-322: Because of this conversion method any comparison of groundwater storage changes from in-situ and GRACE observations will not be independent. This needs to be discussed in some detail. In addition, in the procedure described in Tangdamrongsub et al. (2015) two parameter were used instead of one. Please comment on this difference.*

**R32:** Due to the fact that the estimated scale factor values are in line with the specific yield from the field observations (please also see R33), the bias of the estimated parameter from our approach can be considered small over the Shiyang River Basin. However, we understand reviewer's concern, and therefore one additional paragraph will be added to the conclusion section **lines 809 – 815** as follows:

"The conversion approach between the groundwater head measurement and groundwater storage is proven feasible over the Shiyang River Basin. The approach delivers comparable ranges of scale factor estimates to the specific yield estimated from the field observation. However, it is noted here that the results of the conducted validation might be over-optimistic, since the well data processed with the adopted conversion procedure are not fully independent of assimilated GRACE data. The specific yield from the field observation must be used when available."

Additionally, the difference between 1 and 2 parameters are only the bias (first parameter, "a" parameter in Tangdamrongsub et al. (2015)) becomes very small (~1e-14) when the TWS variation and head variation are considered. Therefore, Eq. (1,2) of Tangdamrongsub et al. (2015) and Eq. (2,3) in the submitted manuscript provide the same result. However, for consistency, we restore the bias term in the revised manuscript as

$$\Delta \text{GWS}_{(\text{GRACE}-\Delta \text{SM})} + e = b + f \cdot \Delta h \qquad (2)$$

$$\Delta \text{GWS}_{\text{in situ}} = \hat{b} + \hat{f} \cdot \Delta h \qquad (3)$$

*L317-318: Please provide a figure with the data and the regression.*

**R33:** The figure of the regression analysis is shown below (Fig. R1). To reduce the redundancy, we do not include Fig. R1 in the manuscript, but instead we will include a discussion of the parameter estimation in the revised manuscript **lines 658 – 667** as follows:

"Yang et al. (2001) showed that the specific yield values obtained from the field measurements over the Shiyang River Basin was between 0.01 and 0.3. Although, the measurement was not conducted at the well stations used in this study, the values obtained can be used as a guidance of the specific yield of the Shiyang River Basin. In this study, the head measurements were converted to storage unit with the approach described in Sect. 4.3.1. The bias term in Eq. (3) was found to be very close to zero, as the variation (mean removed) was used in the regression analysis. The estimated scale factor was 0.23, 0.04, 0.24, 0.25, and 0.32 at W1 – W5, respectively, which was in line with the values obtained from the field measurement."

[Figure]

Figure R1. Regression analysis between GRACE-GLDAS and adjusted well measurements in 5 different locations.

*L320: Why are you using an averaged f value to calculate the groundwater storage for each well? I would have thought that the variations in parameter f should represent local variations in storage parameters of the aquifers. Please explain the reasoning behind this procedure.*

**R34:** The parameter is individually estimated and used for each well location. No average parameter is used. For clarity, we will extend the statement in **lines 313 – 314** as follows:

"… and ΔGWS(GRACE-ΔSM) at each individual location, a bias (b), a scale factor (f) …"

*L451: Please explain how you selected these parameters (e.g. did you use a sensitivity test?).*

**R35:** We selected these parameters based on several previous PCR-GLOBWB studies (e.g. Sutanudjaja et al., 2011, 2014), showing that these selected parameters are indeed the sensitive ones to model simulation results. For clarity, the reference will be added to the revised manuscript.

*L526-527: Change into Figure 10*

**R36:** "Fig. 9" will be changed to "Figure 10" in the revised manuscript.

*L545: "on" instead of "of"*

**R37:** "of" will be changed to "on" in the revised manuscript.

*L549-550: Please provide information on the origin of these parameter values.*

**R38:** The origin of the parameter values is given in Sutanudjaja et al. (2011, 2014), and the reference will be given in the revised manuscript.

Further information related to the origin of parameter values were provided in Appendix A of the submitted manuscript. However, they are removed based on reviewer suggestion (see R21).

The model parameters of PCR-GLOBWB are derived from several globally available datasets that are listed as follows. The Global Land Cover Characteristics Data Base Version 2.0 (GLCC 2.0, http://edc2.usgs.gov/glcc/globe int.php) and and FAO soil maps (1995) were used in order to parameterize the land cover and upper sub-surface properties. For mapping aquifers and estimating the groundwater recession coefficient, the GLobal HYdrogeology MaPS (GLHYMPS) global maps of permeability and porosity (Gleeson et al., 2014), as well as available global digital elevation models (e.g. HydroSHEDS, Lehner et al., 2008) were used. For further explanation about the PCR-GLOBWB model parameterization, the reader is referred to the technical reports (e.g. van Beek and Bierkens, 2009; van Beek, 2008); and other relevant publications (e.g. Sutanudjaja et al., 2011, 2014).

*L543: How do you know that the groundwater store of the Desert Region is small.*

**R39:** We realized that the statement is misleading and we change our statement in **lines 552 – 553** as " …the small amplitude of the groundwater variation of this region …". Small GWSV over the Desert Region is presented in Fig. 10k.

*L553-554: Please explain in greater detail, why higher values of K_sat and lower values of J have led to a smaller amount of water addition.*

**R40:** We realize that the interpretation the amount of water storage in terms of K_sat and J might be misleading as they do not have a linear relationship. Instead, soil water storage capacity (SC, see Table 1) and forcing data have greater impact on the water storage estimate. Note that greater SC value leads to greater amount of water stored in soil layer, and consequently lesser water percolate to the groundwater store. Therefore, we remove the statement related to K_sat and J, and change the analysis to:

"The impact of GRACE DA on different stores was influenced by both the model parameters and the forcing data. The 4 basins have similar soil water storage capacities (see Table 3), which indicates that the basins can store similar amounts of soil water and generate similar amounts of groundwater recharge under the same rainfall conditions. However, the 4 basins received different amounts of rainfall, which resulted in different SMSV and GWSV estimates. For example, the Shiyang River Basin received the greatest amount of rainfall (~ twice of Heihe River Basin), which led to the greatest amount of the SMSV estimate (~1 cm annual amplitude). Such a large amount was also sufficient to percolate into the groundwater layer, resulting in GWSV of ~0.7 cm (see Fig. 10i and Table 2). In contrast, the Desert Region received approximately 3 times less rainfall, which led to a somewhat smaller amount of SMSV (~0.7 cm annual amplitude) and a much smaller amount of GWSV, ~0.2 cm (see Fig. 10g, k).

The above paragraph will be added to the revised manuscript **lines 557 – 571**.

*L599-600: I wonder whether the better agreement with the GRACE DA results is due to (or a least partly due to) the scaling procedure of the piezometer data. Please add a discussion on this.*

**R41:** Due to the fact that the estimated scale factor values are in line with the specific yield from the field observations (please also see R33), the bias of the estimated parameter from our approach can be considered small over the Shiyang River Basin. However, we understand reviewer's concern, and therefore add one additional paragraph into the conclusion of the revised manuscript **lines 809 – 815** as follows:

"The conversion approach between the groundwater head measurement and groundwater storage is proven feasible over the Shiyang River Basin. The scale factor estimates produced with this approach are consistent with the specific yield estimated from the field observations. However, it is noted here that the results of the conducted validation might be over-optimistic, since the well data processed with the adopted conversion procedure are not fully independent of the assimilated GRACE data. The specific yield from the field observation must be used when available."

*L642: Clearly, predictions for G2 were improved to a lesser degree.*

**R42:** We agree with reviewer. For clarity, the statement will be changed to "A lesser improvement was observed at G2", **lines 713** in the revised manuscript.

*L647-648: These are very low amounts of precipitation, indicating very local precipitation events. It would be interesting to see the spatial distribution of these rainfall events and the resulting modelled soil moisture distribution.*

**R43:** The maps of rainfall and SM storage estimates of the discussed events (September 2007, 2008) are shown below (Fig. R2). However, this is beyond the scope of this study, and therefore Fig. R2 is not presented in the manuscript.

[Figure]

Figure R2. Monthly total precipitation (left) and SM storage estimates of September 2007 and 2008. Stream gauge location G2 is also shown.

*L676-678: Why should the SM storage of the Desert Region decrease although precipitation shows an increasing trend? Please discuss.*

**R44:** The discussion will be added to the revised manuscript **lines 749 – 755** as follows:

"In the Desert Region, in contrast to other basins, the minor decreasing TWS trend of -0.1 cm/yr was dominated by loss of SM storage. This was likely caused by inaccurate model parameter calibration over the Desert Region (i.e., too large SC value). Separation of the TWS into groundwater and soil moisture store was likely incorrect. As such, the annual signal in GWS is much less than in SM there. Therefore, GRACE update was mostly attributed to the SM component, so that a groundwater-pumping signature (Jiao et al., 2015) was seen in the SM instead of the GWS layer."

Further discussion of this (and related) issue is also included in the conclusion, **lines 792 - 793**:

"It should be emphasized that GRACE does not fix a technical problem of the hydrological model, but it rather provides information which is not available otherwise."

*L712-714: Until now, there was no indication that groundwater abstraction was considered in the modelling. Please add a description. Why was the groundwater abstraction not considered in the DA?*

**R45:** In this study, the state vector only contains the water storage. As the groundwater abstraction is a parameter of PCR-GLOBWB, it is not included in the state vector. Therefore, the groundwater abstraction is not separately estimated in this study. However, the information of groundwater abstraction is contained in GRACE observation. Once GRACE DA is applied, such information is propagated into the state vector, particularly the groundwater layer. This is clearly seen in the negative trend of updated groundwater estimates. This discussion will be included in the conclusion of the revised manuscript **lines 792 – 797** as follows.

"It should be emphasized that GRACE does not fix a technical problem of the hydrological model, but rather it provides information, which is not available otherwise. Note that, in principle, the model may predict any long-term behaviour of water storage, but that information should be brought in "by hand" (e.g., via the groundwater abstraction parameter). As soon as that information is not available, reliable long-term predictions on the basis of hydrological modelling alone are conceptually impossible."

*L734-735: See comment above. Would it be possible to update the groundwater abstraction parameter?*

**R46:** Yes, it is possible to update the model parameter together with the state vector. We will consider reviewer's suggestion in the future work.

*L744: Please provide quantitative information on groundwater abstraction.*

**R47:** As the groundwater abstraction is not estimated by our GRACE DA approach, we do not quantify the amount of groundwater abstraction in this study. The groundwater abstraction can be quantified when the parameter is estimated together with the state vector.

*Fig. 1. Include all symbols in the figure caption (crosses). Since color is used, the river networks could also be added (1b).*

**R6:** The symbol will be added to Fig. 1 caption of the revised manuscript **lines 1130 – 1131** as " …the locations of considered groundwater wells (x) and river stream gauges (+)." The river network will also be added to Fig. 1b.

*p6, l208/209. Please, explain 'the sum of different states'. What are e.g. '4 interception' states?*

**R7:** TWS variation is computed from the sum of 27 different water storage components (layers), which are 8 soil moisture layers, 2 groundwater layers, 4 interception layers, 8 snow layers, 4 inundated top water layers, and 1 surface water layer. For clarity, we revise the statement in the revised manuscript **lines 197 – 199** to:

"… the total water storage (TWS) is computed as the sum of 27 different water storage components: 8 soil moisture layers, 2 groundwater layers, 4 interception layers, 8 snow layers, 4 inundated top water layers, and 1 surface water layer.".

*p9, l331ff. What exactly was done with the NDVI values? Was the growing season length determined as the period above and below 0.2? If it was only used for visualization in Fig. 14, the section can be shortened to a couple of lines.*

**R8:** NDVI and GWS variation were analysed together to determine if the growing season was being extended beyond the limited rainy period through groundwater extraction for irrigation. The reviewer is correct in that the growing season length is determined as the period above ~0.2. In the revised manuscript, we remove a few statements in Sect. 4.4.3 to make the section more concise.

*Fig. 14a. Is the GW head relative to amsl? What is the depth to the surface?*

**R9:** Yes, the measurement is relative to the mean sea level. For clarity, we will add an additional statement to the revise manuscript **line 295**:

"… form of piezometric heads (relative to the mean sea level), …"

The depth from to the surface is not available from the data provider, and therefore we cannot provide the value here.

**R4**: Please see R1

*Eq. (8): Since both error covariance matrices (from the model and the observations) have a rank-defect due to (1) the fact that usually the number of model states is much larger than the number of model ensemble members and (2) GRACE cannot actually resolve TWS changes on a 0.5x0.5 degree grid, the inverse in Eq. (8) does not exist. This should be pointed out and a reference to sections 5.2.1 and 5.2.2 might be provided that describe how the authors deal with this issue.*

**R5**: Please see R2

*l. 251: GRACE observations are highly correlated on such a fine spatial resolution (similar to the above comment). Did the authors investigate this? Was this the reason to use a maximum correlation length for the observation error covariance matrix?*

**R6**: Reviewer is correct. In our covariance computation (described in Sect. 5.2.2), the localization function with correlation length similar to the Gaussian smoothing used was applied. The localization helps to improve the matrix stability and we investigated this by checking the rank and condition number of the matrix as explained in R2.

*l. 414-415: If I understand it correctly, the error correlation length is set to 250 km and TWS changes outside of this radius are assumed to not be correlated to the center grid cell. Is this reasonable? It would be helpful to investigate the correlations of points with longer distances to verify this choice. Does the "local" error covariance matrix have a full rank?*

**R7**: As the observation error variance-covariance matrix is derived based on the application of 250 km filter radius, the correlation error at distance beyond 250 km (correlation length) does not have a crucial impact on the result. In the submitted manuscript, we demonstrate the error characteristic in Fig. 8b. From the figure, the correlation reduces significantly beyond the correlation length. Additionally, the error variance-covariance matrix derived based on our method has a full rank.

*Fig. 7: In the main text (l. 414-415), it is explained that a correlation length of 250 km is used (approx. four to five 0.5x0.5 degree (~50kmx50km at the equator) grid cells in each direction from the center grid cell). In Fig. 7, it is shown that only the neighboring grid cells are considered. Please clarify.*

**R8**: Reviewer is correct. We realized that the figure caption was not explained clearly. To clarify this, we add an additional description in the figure caption **lines 1158 – 1160** as follows:

"The graphic demonstrates the case of 1 pixel (0.5 degree) correlation distance. The boundary stretches farther for larger correlation distance."

*l. 419: Since the neighboring 0.5x0.5 degree grid cells are highly correlated, it is not reasonable - based on the GRACE error characteristics - to apply the EnKF without spatial error correlations on such a fine scale. A statement would be helpful to the reader.*

**R9**: We thank reviewer for the recommendation, the statement will be added to the revised manuscript lines 399 – 400 as follows:

"Spatial correlations of model errors and observation errors were also taken into account in view of the fact that the latter are highly correlated at neighbouring $0.5^{\circ} \times 0.5^{\circ}$ grid cells."

*l. 726-727: But: The authors do not use the full error covariance matrix as directly calculated from the observations. Instead a maximum correlation length of 250 km is*

*assumed, and thus a part of the information within the full error covariance matrix is neglected. Therefore, the statement might be misleading.*

**R10**: We thank for reviewer comment. To clarify this, we will modify the statement in the revised manuscript **lines 830 – 832** as follows:

"…this is a reasonable price to pay as deriving the error variance-covariance matrix from the full (and only full) error covariance matrix noticeably improves the results of GRACE data assimilation."

*l. 90-91: That seems to be incorrect. Zaitchik et al. (2008) used an ensemble Kalman smoother (EnKS) approach to partition the monthly update increment (based on comparing monthly means of modeled and observed TWS changes) equally to each day of the month. GRACE TWS changes are only assimilated once per month and not every 10 days.*

**R11**: We thank for reviewer comment. The statement will be corrected in the revised manuscript **lines 197 – 199** as follows:

"…using a monthly observation value and distributing the update as daily increments (Zaitchik et al., 2008; Forman et al., 2012; Girotto et al. 2016)."

*l. 95: This work adapts the method as proposed in Zaitchik et al. (2008) to a snow-dominated basin.*

**R12**: Please see R11.

*l. 98: Please also consider the disadvantage of computational costs: The method has some computational drawback since the model has to be evaluated twice over the same month.*

**R13**: We thank for reviewer suggestion. The additional sentence will be added in the revised manuscript **lines 102 – 103** as follows:

"The only price to pay is the additional computational cost of running the model twice for the same month."

*l. 106: In Forman et al. (2013), the authors did not use correlated errors for the data assimilation. They investigated for which spatial resolution errors of GRACE TWS changes might be considered as uncorrelated. According to these investigations, they assumed white noise for (sub-)basin averaged TWS changes from GRACE.*

**R14**: We agree with reviewer. Forman et al. (2013) will be removed in this context to avoid the confusion.

*l. 89-95: In this work, the authors performed an analysis of introducing the update increments completely at the beginning of a month, the end of a month or equally distributed over all days of a month. This is worth to be mentioned along with the other citations.*

**R15**: We thank for reviewer suggestion. Girotto et al. (2016) will be cited in the revised manuscript.

*l. 39-40 and l. 106-108: A first analysis of assessing the effect of considering or neglecting spatial error correlations of GRACE TWS changes was performed in Schumacher et al. (2016) in form of a synthetic experiment, for which one of the authors of this HESSD manuscript was the editor and should therefore be very familiar with the work. It seems that the paper is methodologically the closest related to the analysis presented here and,*

*therefore, should be cited and discussed. Findings should be compared to the findings in the published paper.*

**R16**: At the time this study was conducted, Schumacher et al. (2016) was not published, therefore we conducted the analysis independently based on our method (proposed in this HESSD paper). However, we thank reviewer for the recommendation, and Schumacher et al. (2016) will be cited in the revised manuscript.

*l. 577: This was also seen and discussed in Schumacher et al. (2016). The authors should compare their results with the findings in this paper, since the objective of both papers is to understand the effect of considering spatial error correlations of GRACE TWS changes on hydrological data assimilation results.*

**R17**: We thank for reviewer suggestion. The suggestion will be considered in the revised manuscript.

*l. 715-718: The authors should add something like "in agreement with the recommendation in Schumacher et al. (2016)."*

**R18**: We thank for reviewer suggestion. The statement will be given in the revised manuscript **lines 823 – 824**.

*l. 719-724: The findings in the HESSD manuscript allow for a clearer conclusion on improvements when error correlations of GRACE TWS changes are taken into account. What might be the reason for this? Differences in the study set up? Localization of model / observation error covariance matrices?*

**R19:** The improvement is mainly due to a better representation of GRACE information in the EnKF. Ignoring error correlations in the DA led to an over-fit of the results to the observations, which led to less accurate state estimates. These explanation will be presented in the revised manuscript **lines 824 – 828** as follows:

"We explain this finding by the fact that GRACE errors at the neighbouring $0.5^{o}$x$0.5^{o}$ grid cells are highly correlated. As such, the simultaneous consideration of GRACE data at multiple neighbouring cells does not reduce data noise, as it would be the case if noise were white. In other words, the white-noise assumption may severely overestimate the information content of GRACE data."

*l. 729: A reference to Schumacher et al. (2016) would strengthen this statement, since the HESSD manuscript is not the only study that concludes a benefit / more realistic GRACE data assimilation approach if implementing GRACE error correlations.*

**R20**: We thank for reviewer for the suggestion, Schumacher et al. (2016) will be cited in the relevant context.

*l. 752-753: Schumacher et al. (2016) should be added to the list of references.*

**R21**: Schumacher et al. (2016) will be added to the list of references.

*l. 755: Alternative methods have been investigated in Schumacher et al. (2016), namely a square root analysis scheme (SQRA) and the singular evolutive interpolated Kalman filter (SEIK). Especially the application of the SEIK filter showed promising results. A citation would support the authors expectation that alternative methods, e.g. the particle filter, would improve the data assimilation performance.*

**R22**: We thank the review for the suggestion.  We will consider this in the revision.

*l. 583: "truth", i.e. to the independent measurements of individual water compartments. These measurements are also subject to uncertainties and not "true" values.*

**R23**: To avoid the confusion, the statement will be changed to "Validating against the in situ groundwater and streamflow data will quantitatively reveal the performance of each approach". This is given in lines 630 – 631 of the revised manuscript.

*l. 756: "true" -> better "full" (true is difficult since often unknown / poorly known)*

**R24**: "true" will be changed to "realistic".

[revised manuscript text omitted]